# Meta Continual Learning Revisited: Implicitly Enhancing Online Hessian Approximation via Variance Reduction

**Yichen Wu**[1,2][*], **Long-Kai Huang**[2][†], **Renzhen Wang**[3], **Deyu Meng**[3,4], **Ying Wei**[1,5][†]
[1]City University of Hong Kong, [2]Tencent AI Lab, [3]Xi'an Jiaotong University,
[4]Pazhou Laboratory (Huangpu), [5]Nanyang Technological University

## Abstract

Regularization-based methods have so far been among the *de facto* choices for continual learning. Recent theoretical studies have revealed that these methods all boil down to relying on the Hessian matrix approximation of model weights. However, these methods suffer from suboptimal trade-offs between knowledge transfer and forgetting due to fixed and unchanging Hessian estimations during training. Another seemingly parallel strand of Meta-Continual Learning (Meta-CL) algorithms enforces alignment between gradients of previous tasks and that of the current task. In this work we revisit Meta-CL and for the first time bridge it with regularization-based methods. Concretely, Meta-CL implicitly approximates Hessian in an online manner, which enjoys the benefits of timely adaptation but meantime suffers from high variance induced by random memory buffer sampling. We are thus highly motivated to combine the best of both worlds, through the proposal of Variance Reduced Meta-CL (VR-MCL) to achieve both timely and accurate Hessian approximation. Through comprehensive experiments across three datasets and various settings, we consistently observe that VR-MCL outperforms other SOTA methods, which further validates the effectiveness of VR-MCL[1].

## 1 Introduction

Continual learning (CL) is a promising approach to constructing learning systems that can continuously process information streams, adapt to changing environments, and acquire new knowledge while consolidating and retaining previously learned knowledge (Parisi et al., 2019). In contrast to traditional supervised learning, which trains on independent and identically distributed (i.i.d) samples, continual learning involves models trained on non-stationary data distributions, where different classification tasks are presented sequentially. The violation of the i.i.d. assumption in CL can lead to catastrophic forgetting, causing a significant drop in test performance on previously learned tasks (French, 1999; McClelland et al., 1995; McCloskey & Cohen, 1989).

In recent years, numerous methods in the field of CL have been proposed with the aim of tackling the catastrophic forgetting problem. Regularization-based methods (Kirkpatrick et al., 2017; Ritter et al., 2018; Huszár, 2017; Zenke et al., 2017), being an important branch of these methods, aim to discern and retain the important parameter weights associated with prior tasks so as to sustain their performance. Specifically, they assess the importance of different parameter weights through the second-order Hessian of previous tasks computed at the end of each task training. The differences among these regularization-based methods primarily reside in their distinct approaches to approximating the Hessian. For example, EWC (Kirkpatrick et al., 2017) and Online-EWC (Huszár, 2017) employ the diagonal Fisher information matrix, whereas Kronecker factored Laplace approximation (KFLA) (Ritter et al., 2018) utilizes a more accurate Kronecker factorization approximation, considering off-diagonal elements. Similarly, Intelligent Synapses (IS) (Zenke et al., 2017) adopts an approximate diagonal matrix, but its elements take into account the importance of a task throughout the entire training trajectory. Despite these methods striving to improve the Hessian approximation

---

[*]Part of the work was done when the author interned in Tencent AI Lab (wuyichen.am97@gmail.com).
[†]Corresponding author: Long-Kai Huang (hlongkai@gmail.com) and Ying Wei (ying.wei@ntu.edu.sg)
[1]Code is available at https://github.com/WuYichen-97/Meta-CL-Revised

to preserve important weights for previous tasks, their estimated Hessian matrices of previous tasks remain unchanged and fail to update during subsequent training. As the model parameters gradually deviate from the point at which the Hessian was initially computed, the unaltered estimated Hessian progressively loses accuracy due to the increasing truncation error of the Taylor expansion.

Different from regularization-based methods that *explicitly* approximate the second-order Hessian at the end of training for each task, as will be detailed in Sec. 3, Meta-Continual Learning (Meta-CL) methods utilize the second-order Hessian information *implicitly* through the hypergradient obtained via bi-level optimization (Riemer et al., 2019; Gupta et al., 2020; Von Oswald et al., 2021). The computation of hypergradient involves a batch of samples drawn from the memory buffer, which stores examples from previous tasks, enabling Meta-CL to leverage up-to-date second-order information from previous tasks. However, it is noteworthy that the utilization of samples drawn from the memory buffer may not offer a complete representation of previous tasks, and may even lack data pertaining to certain tasks. This can result in the presence of erroneous information in the Hessian estimation. As a consequence, there may be considerable variation in the implicitly estimated Hessian, ultimately leading to a decline in performance for previous tasks.

To leverage the benefits of Meta-CL, specifically the timely updating of second-order Hessian information, while mitigating the negative impact of variation, we propose a momentum-based **V**ariance-**R**educed **M**eta-**CL** (**VR-MCL**). Through our theoretical analysis, we demonstrate that the integration of the momentum-based variance-reduced technique into Meta-CL is equivalent to imposing a penalty on its online estimated Hessian. This penalty effectively suppresses excessive model updates, thereby better preserving crucial weights for previous tasks and their performance. Our main contributions are summarized as follows:

- We introduce a new perspective on Meta-CL by characterizing it as a technique that approximates the Hessian matrix, thereby establishing a connection between Meta-CL and regularized methods.
- To address the issue of high variance in Meta-CL, we propose Variance-Reduced Meta-CL (VR-MCL), which incorporates hypergradient variance reduction technique into Meta-CL.
- Theoretically, we demonstrate that our proposed momentum-based hypergradient variance reduction technique is equivalent to the inclusion of an effective penalty term within the implicitly estimated Hessian. Additionally, we provide the regret bound of VR-MCL to further explain its superior performance in the context of continual learning.
- We conduct comprehensive experiments covering three datasets and multiple continual learning settings, providing empirical evidence for the effectiveness of the proposed algorithm.

## 2 THE UNIFIED FRAMEWORK OF REGULARIZATION-BASED CL METHODS

In this section, we will first derive the unified iterative updating framework used by regularization-based algorithms following the analysis of Yin et al. (2020) and then show the impact of Hessian approximation accuracy on the performance of these algorithms.

**Analysis of Regularization-based methods.** Continual learning considers a sequence of $N$ tasks $[\mathcal{T}^1, \mathcal{T}^2, \cdots, \mathcal{T}^N]$, where the $i$-th task consists of $N^i$ samples, i.e., $\mathcal{T}^i = (\mathbf{X}^i, \mathbf{Y}^i) = \{(\mathbf{x}_n^i, y_n^i)\}_{n=0}^{N^i}$. Here, $\mathcal{T}^j$ represents the current training task, $\mathcal{T}^{[1:j]}$ refers to all $j$ tasks the model have seen so far, and $\mathcal{L}^i(\theta)$ denotes the empirical risk of the parameter $\theta$ on $\mathcal{T}^i$. The objective of CL is to learn a model with parameters $\theta$ that minimizes the empirical risk of all seen tasks $\mathcal{T}^{[1:j]}$, i.e., $\frac{1}{j}(\sum_{i=1}^{j} \mathcal{L}^i(\theta))$. However, in the context of regularization-based CL, we have no access to the samples of previously learned tasks $\mathcal{T}^{[1:j-1]}$. The model parameters $\theta$ cannot be directly optimized to minimize the corresponding empirical risks, i.e., $\sum_{i=1}^{j-1} \mathcal{L}^i(\theta)$. Therefore, regularized CL methods choose to approximate the empirical risk of previous tasks during training on $\mathcal{T}^j$ (Yin et al., 2020). Concretely, in the simplest case where $j=2$, when training on $\mathcal{T}^2$, the objective of regularization-based methods is to minimize $\frac{1}{2}(\mathcal{L}_1^{\mathrm{prox}} + \mathcal{L}^2)$, where $\mathcal{L}_1^{\mathrm{prox}}(\theta)$ is an approximation of $\mathcal{L}^1(\theta)$ derived through a Taylor expansion at $\hat{\theta}^1$ as $\mathcal{L}_1^{\mathrm{prox}} = \mathcal{L}^1(\hat{\theta}^1) + (\theta - \hat{\theta}^1)^\top \nabla \mathcal{L}^1(\hat{\theta}^1) + \frac{1}{2}(\theta - \hat{\theta}^1)^\top \nabla^2 \mathcal{L}^1(\hat{\theta}^1)(\theta - \hat{\theta}^1)$. Here, $\hat{\theta}^1$ represents the model parameters at the end training of $\mathcal{T}^1$. For a more general setting where $j > 2$, we can approximate the empirical loss of the previous $(j-1)$ tasks by,

$$\mathcal{L}_{j-1}^{\mathrm{prox}}(\theta) = \sum_{i=1}^{j-1} \underbrace{\mathcal{L}^i(\hat{\theta}^i)}_{(a)} + \underbrace{(\theta - \hat{\theta}^i)^\top \nabla_\theta \mathcal{L}^i(\hat{\theta}^i)}_{(b)} + \underbrace{\frac{1}{2}(\theta - \hat{\theta}^i)^\top \mathbf{H}^i(\theta - \hat{\theta}^i)}_{(c)}, \tag{1}$$

Table 1: Summary of continual learning methods under the unified iterative update rule with various instantiations of Hessian approximation. Here $\mathbf{F}$ and $\mathbf{F}_D$ are the full and diagonal Fisher information matrices, respectively. $\mathbf{H}_D$ denotes diagonal approximation of the Hessian matrix, and $\mathbf{H}_M$ and $\mathbf{H}_{\mathrm{VR}}$ are the online Hessian estimated on previous tasks and its regularized version, respectively.

| Method | Hessian Approximation | | | Iterative Update Rule |
|---|---|---|---|---|
| | Off-diagonal elements | Online update | Penalty | |
| EWC / On-EWC | ✗ | ✗ | ✗ | $\theta := \theta - \alpha(\sum_{i=1}^{j-1} \mathbf{F}_D^i)^{-1}\nabla_\theta \mathcal{L}^j(\theta)$ |
| KFLA | ✓ | ✗ | ✗ | $\theta := \theta - \alpha(\sum_{i=1}^{j-1} \mathbf{F}^i)^{-1}\nabla_\theta \mathcal{L}^j(\theta)$ |
| IS | ✗ | ✓ | ✗ | $\theta := \theta - \alpha(\sum_{i=1}^{j-1} \mathbf{H}_D^i)^{-1}\nabla_\theta \mathcal{L}^j(\theta)$ |
| La-MAML | ✓ | ✓ | ✗ | $\theta := \theta - \alpha(\mathbf{H}_M^j)^{-1}\nabla_\theta \mathcal{L}^j(\theta)$ |
| VR-MCL | ✓ | ✓ | ✓ | $\theta := \theta - \alpha(\mathbf{H}_{\mathrm{VR}}^j)^{-1}\nabla_\theta \mathcal{L}^j(\theta)$ |

where $\hat{\theta}^i$ and $\mathbf{H}^i = \nabla_\theta^2 \mathcal{L}^i(\hat{\theta}^i)$ denote the model parameter at the end of training $\mathcal{T}^i$ and the Hessian matrix, respectively. For the concrete analysis of $\mathcal{L}_{j-1}^{\mathrm{prox}}(\theta)$ in Eqn. (1), the first term (a) is not related to $\theta$ and thus can be discarded. Furthermore, since the training process at the end of each task is usually converged, the gradient $\nabla \mathcal{L}_i(\hat{\theta}_i)$ is near zero and the term (b) can be ignored in practice. Therefore, the empirical loss approximation $\mathcal{L}_{j-1}^{\mathrm{prox}}(\theta)$ is almost equivalent to approximating the Hessian matrix in item (c). Based on this approximation and the analysis, we can derive the iterative update rule of regularized methods in Proposition 1. (See Appendix A.1 for proof.)

**Proposition 1.** *In regularization-based continual learning, if the model parameter $\theta$ is searched within the neighborhood set $\cup_{i=1}^{j-1}\mathcal{N}^i$ with $\mathcal{N}^i = \{\theta : d(\theta, \hat{\theta}^i) < \delta^i\}$, then the iterative update rule of $\theta$ approximately is,*

$$\theta :\approx \theta - \alpha(\mathbf{H}^1 + \mathbf{H}^2 + \cdots + \mathbf{H}^{j-1})^{-1}\nabla_\theta \mathcal{L}^j(\theta). \tag{2}$$

**The Effect of Hessian in Regularization-based CL.** Let $\mathbf{H} = \sum_{i=1}^{j-1} \mathbf{H}^i$ denote the Hessian part in Eqn. (2). This update rule implies that the Hessian $\mathbf{H}$ plays a crucial role in the optimization process. Specifically, by multiplying the inverse of $\mathbf{H}$ with the gradient $\nabla_\theta \mathcal{L}^j(\theta)$, the gradient is suppressed along the large curvature directions, i.e., the directions along the eigenvectors corresponding to large eigenvalues of $\mathbf{H}$, and amplified along the small curvature directions. In regularization-based CL algorithms, the large curvature directions of the Hessian align with the important weights storing the knowledge of the previous tasks, which suppresses the update scale to reduce forgetting, while the small curvature directions facilitate updating the model to fit the current task.

**Regularization-based CL in a Unified Framework.** Given the critical role of the Hessian in model updates, various regularization-based methods primarily focus on enhancing their approximation of the Hessian. By integrating Proposition 1, we encapsulate existing regularization-based CL methods within a unified iterative update framework as summarized them in Table 1. Concretely, EWC (Kirkpatrick et al., 2017) and On-EWC (Huszár, 2017) utilize the diagonal Fisher information matrix $\mathbf{F}_D^i$ to approximate the $\mathbf{H}^i$ in Eqn. (2). In contrast, Intelligent Synapses (IS) (Zenke et al., 2017) employs a diagonal matrix $\mathbf{H}_D^i$, with elements signifying the importance of a task throughout the entire training trajectory. To enhance the approximation of the Hessian, Kronecker factored Laplace approximation (KFLA) (Ritter et al., 2018) opts to use the Kronecker factorization approximation $\mathbf{F}^i$, taking into account off-diagonal elements, to approximate $\mathbf{H}^i$.

## 3 REVISITING META-CL FROM A HESSIAN APPROXIMATION PERSPECTIVE

In this section, we revisit the Meta-CL from a Hessian approximation perspective. Specifically, while regularization-based methods explicitly approximate the second-order Hessian, Meta-CL utilizes second-order information through the computation of hypergradient. Consequently, it can also be scrutinized within the iterative update framework presented in Eqn. (2).

**Meta-Continual Learning (Meta-CL).** As one of the earliest works of Meta-CL, Riemer et al. (2019) propose a meta-learning formulation for minimizing the empirical risk of all seen tasks $\mathcal{T}^{[1:j]}$. In this approach, the data in $\mathcal{T}^{[1:j-1]}$ are partially accessible via a memory buffer $\mathcal{M}$, which stores

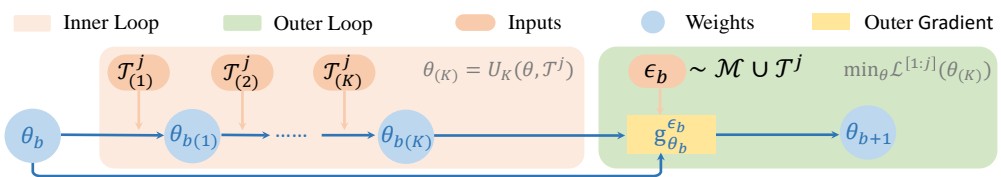

Figure 1: Iterative update process of Meta-CL for the $b$-th iteration. The notation $\mathcal{T}^j_{(k)}$ means samples drawn from $\mathcal{T}^j$, $\mathcal{M}$ refers to the memory buffer and $\mathbf{g}^{\epsilon_b}_{\theta_b}$ is the update gradient.

data from previous tasks. Concretely, the optimization problem is formulated as,

$$\min_\theta \ \mathcal{L}^{[1:j]}(\theta_{(K)}) \quad s.t. \ \theta_{(K)} = U_K(\theta; \mathcal{T}^j),$$

$$\text{where} \ \ U_K(\theta; \mathcal{T}^j) = \overbrace{U \circ \cdots \circ U \circ U}^{K \text{ inner steps}}(\theta_{(0)}; \mathcal{T}^j_{(0)}), \quad U(\theta_{(k)}; \mathcal{T}^j_{(k)}) = [\theta - \beta \nabla_\theta \mathcal{L}^j_{(k)}(\theta)]\big|_{\theta = \theta_{(k-1)}}, \quad (3)$$

the $\mathcal{L}^{[1:j]}(\theta_{(K)})$ characterizes the empirical risk of $\theta_{(K)}$ on all $j$ tasks $\mathcal{T}^{[1:j]}$ which in practice is estimated on $\mathcal{T}^j$ and the data in memory buffer $\mathcal{M}$, $\theta_{(k)}$ denotes the parameters of the $k$-th **inner step** of learning for task $\mathcal{T}^j$, and $U(\cdot)$ means one-step SGD update in the inner loop optimization.

We illustrate the iterative update process for the model parameters $\theta$ for the $b$-th iteration, denoted by $\theta_b$, in Fig. 1. In the inner loop, the model is initialized with $\theta_{b(0)} = \theta_b$ and iteratively updated via $K$ steps of SGD updates, where $\mathcal{T}^j_{(k)}$ is the data randomly sampled from $\mathcal{T}^j$ in the $k$-th step. Following the acquisition of $\theta_{b(K)}$, in the outer loop, we randomly sample data $\epsilon_b$ from $\mathcal{T}^j$ and the memory buffer $\mathcal{M}$. Then we compute the loss $\mathcal{L}^{[1:j]}(\theta_{b(K)})$ and obtain the hypergradient w.r.t. $\theta_b$ as $\mathbf{g}^{\epsilon_b}_{\theta_b} := \partial \mathcal{L}(\theta_{b(K)}; \epsilon_b)/\partial \theta_b$. We finally update $\theta_b$ using the hypergradient. It should be noted that different Meta-CL methods typically employ different approximations of the update gradient $\mathbf{g}^{\epsilon_b}_{\theta_b}$. For instance, MER (Riemer et al., 2019) utilizes a first-order approximation, while La-MAML (Gupta et al., 2020), following MAML (Finn et al., 2017), directly computes the hypergradient.

**Revisiting from the Hessian approximation perspective.** To solve the bi-level optimization problem as shown in Eqn. (3), we need to compute the hypergradient of $\mathcal{L}^{[1:j]}(\theta_{(K)})$ *w.r.t.* $\theta$, i.e.,

$$\frac{\partial \mathcal{L}^{[1:j]}(\theta_{(K)})}{\partial \theta} = \frac{\partial \mathcal{L}^{[1:j]}(\theta_{(K)})}{\partial \theta_{(K)}} \frac{\partial \theta_{(K)}}{\theta},$$

where the Taylor approximation of the first term around $\theta$ is,

$$\frac{\partial \mathcal{L}^{[1:j]}(\theta_{(K)})}{\partial \theta_{(K)}} = \nabla_{\theta_{(K)}} \mathcal{L}^{[1:j]}(\theta_{(K)}) \approx \nabla_{\theta_{(K)}} \mathcal{L}^{[1:j]}(\theta) + \mathbf{H}^j_M(\theta_{(K)} - \theta) + (\theta_{(K)} - \theta)^T \otimes \mathbf{T} \otimes (\theta_{(K)} - \theta).$$

Note that $\mathbf{H}^j_M = \nabla^2_{\theta_{(K)}} \mathcal{L}^{[1:j]}(\theta)$ and $\mathbf{T}$ denote the Hessian matrix and the third-order symmetric tensor, respectively, and $\otimes$ represents the Kronecker product. Through the above approximation and assuming a single inner step in optimizing $\mathcal{T}^j$ (i.e., $K=1$) for simplicity, we have Proposition 2.

**Proposition 2.** *For Meta-CL with single inner step adaption, suppose that $\theta_{(K)}$ is located in the $\epsilon$-neighborhood $\mathcal{N}(\theta^*, \epsilon)$ of the optimal model parameter $\theta^* = \mathrm{argmin}_\theta \mathcal{L}^{[1:j]}(\theta_{(K)})$, $\mathcal{L}$ is $\mu$-smooth, and $\beta < \sqrt{\delta / |\nabla_\theta \mathcal{L}^j(\theta) - (\nabla_\theta \mathcal{L}^j(\theta))^2|}$ where $\delta$ is a small number. Let $\alpha = \beta^2$, then the iterative update rule approximately is,*

$$\theta := \approx \theta - \alpha (\mathbf{H}^j_M)^{-1} \nabla_\theta \mathcal{L}^j(\theta).$$

We defer the proof and analysis with $K$-step ($K > 1$) adaption to Appendix A.2. Proposition 2 demonstrates that the use of hypergradient by Meta-CL is equivalent to implicitly employing the Hessian ($\mathbf{H}^j_M$) and adheres to the unified iterative update rule of regularization-based techniques. By comparing Proposition 2 and Proposition 1, we can deduce that Meta-CL utilizes the samples drawn from $\mathcal{M}$ to implicitly compute Hessian $\mathbf{H}^j_M$ so as to approximate $(\mathbf{H}^1 + \mathbf{H}^2 + ... + \mathbf{H}^{j-1})$ in Eqn (2). In contrast to regularized methods that estimate and fix each individual $\mathbf{H}^i$, Meta-CL implicitly computes one Hessian $\mathbf{H}^j_M$ enabling it to have the up-to-date second-order information.

Despite the advantages of Meta-CL in utilizing up-to-date Hessian information via $\mathcal{M}$, it can also introduce erroneous information. For example, the samples drawn from $\mathcal{M}$ may not adequately

represent previous tasks or may lack instances from specific tasks. This inadequacy may lead to the estimated Hessian mistakenly having a small curvature direction along important weights of these previous tasks, which in turn fails to suppress updates along this direction. Consequently, this could result in the loss of previous knowledge from those tasks and lead to forgetting.

## 4 VARIANCE REDUCTION ON META-CL

To tackle the issue of high variance in Meta-CL caused by the random sampling strategy in the memory buffer, we propose Variance-Reduced Meta-CL (VR-MCL) that effectively reduces the variance in the hypergradient of Meta-CL. Our theoretical analysis demonstrates that this variance reduction in the hypergradient is equivalent to the addition of regularization to the implicitly estimated Hessian matrix, resulting in a decrease in the variance of the Hessian matrix and an improvement in its estimation accuracy. Moreover, we also provide a theoretical guarantee of the regret bound.

### 4.1 VARIANCE-REDUCED META-CL (VR-MCL)

VR-MCL aims to reduce the variance of its implicitly estimated Hessian by diminishing the variance of its hypergradient. While there are various variance reduction approaches, such as SAG (Gower et al., 2020), SAGA (Defazio et al., 2014), and SVRG (Johnson & Zhang, 2013) for convex optimization as well as the non-convex variants (Allen-Zhu & Hazan, 2016; Nguyen et al., 2017; Fang et al., 2018), directly applying these methods to online continual learning is challenging due to the requirement of computing the full-batch gradient for all samples in a task. However, in the online continual learning setting, samples from a task are received in mini-batches, and the full sample set remains inaccessible, which poses the challenge to the implementation of these techniques.

On this account, we propose a momentum-based variance-reduced Meta-Continual Learning (Meta-CL) method, inspired by previous works Cutkosky & Orabona (2019); Khanduri et al. (2021). Instead of computing the full-batch gradient which is impractical for online CL, our method, VR-MCL, retains the parameters from the previous iterative step. The $b$-th iteration of VR-MCL is illustrated in Fig. 2, where $\hat{\mathbf{g}}_{\theta_{b-1}}^{\epsilon_{b-1}}$ represents the momentum component and $\mathbf{g}_{\theta_{b-1}}^{\epsilon_b}$ and $\mathbf{g}_{\theta_b}^{\epsilon_b}$ denote the hypergradients on previous $\theta_{b-1}$ and $\theta_b$, respectively, computed using data $\epsilon_b$. In VR-MCL, the final update gradient with reduced variance is shown in Eqn. (4), where $r$ represents the momentum ratio. For clarity, we present the pseudo-codes of the algorithm in Appendix E.

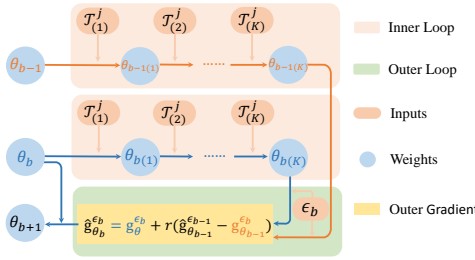

Figure 2: The iterative update process of VR-MCL for the $b$-th iteration, where $\hat{\mathbf{g}}_{\theta_b}^{\epsilon_b}$ is the final update gradient to obtain the $\theta_{b+1}$.

$$\hat{\mathbf{g}}_{\theta_b}^{\epsilon_b} = \mathbf{g}_{\theta_b}^{\epsilon_b} + r(\hat{\mathbf{g}}_{\theta_{b-1}}^{\epsilon_{b-1}} - \mathbf{g}_{\theta_{b-1}}^{\epsilon_b}). \tag{4}$$

To comprehend why Eqn. (4) delivers a smaller gradient variance, we initially examine $\Delta_b := \hat{\mathbf{g}}_{\theta_b}^{\epsilon_b} - \mathrm{G}_{\theta_b}$. This term measures the error incurred when we use $\hat{\mathbf{g}}_{\theta_b}^{\epsilon_b}$ as the gradient instead of the true, but unknown full-batch gradient direction $\mathrm{G}_{\theta_b}$. Through demonstrating that the term $\mathbb{E}[\|\Delta_b\|^2]$ decreases over time, we can show the effectiveness of the VR-MCL in variance reduction Cutkosky & Orabona (2019). Substituting Eqn. (4) into $\Delta_b$, we can obtain the following expression,

$$\Delta_b = r\Delta_{b-1} + (1-r)(\mathbf{g}_{\theta_b}^{\epsilon_b} - \mathrm{G}_{\theta_b}) + r(\mathbf{g}_{\theta_b}^{\epsilon_b} - \mathbf{g}_{\theta_{b-1}}^{\epsilon_b} - (\mathrm{G}_{\theta_b} - \mathrm{G}_{\theta_{b-1}})). \tag{5}$$

The second term $(1-r)(\mathbf{g}_{\theta_b}^{\epsilon_b} - \mathrm{G}_{\theta_b})$ can be modulated by adjusting the ratio of $r$, and the third term $\mathbf{g}_{\theta_b}^{\epsilon_b} - \mathbf{g}_{\theta_{b-1}}^{\epsilon_b} - (\mathrm{G}_{\theta_b} - \mathrm{G}_{\theta_{b-1}})$ is expected in the order of $O(\|\theta_b - \theta_{b-1}\|) = O(\alpha\|\hat{\mathbf{g}}_{\theta_{b-1}}^{\epsilon_{b-1}}\|)$ given the $\mu$-smooth loss function $\mathcal{L}$. Thus, we have $\|\Delta_b\| = r\|\Delta_{b-1}\| + S$ where $S$ is a number that influenced by $r$ and the learning rate $\alpha$. As $\|\Delta_b\|$ keeps decreasing until it reaches $S/(1-r)$, choosing appropriate values of $r$ and $\alpha$ that render $S/(1-r)$ as small as possible ensures variance reduction as we expect.

**Remark.** As previously analyzed, most variance reduction methods, such as SAG (Gower et al., 2020), SVRG (Johnson & Zhang, 2013) and their nonconvex version, are not applicable in online CL settings. Therefore, we design the momentum-based variance reduction technique specifically for Meta-CL. We also discuss other potentially viable variance reduction methods in Sec. 5.

## 4.2 THEORETICAL ANALYSIS OF VR-MCL.

In this subsection, we will demonstrate that the reduction of hypergradient variance by VR-MCL is equivalent to imposing a penalty on its implicitly estimated Hessian. As a result, it effectively mitigates the adverse effects of significant Hessian variation, as detailed in Proposition 3. Additionally, we also provide the regret bound of VR-MCL in Theorem 1 to further explain its effectiveness.

**Proposition 3.** *Assume that the batch size for inner step adaptation is sufficiently large and the b-th step trains on $\mathcal{T}^j$. Let $\mathbf{H}^j_{M_b} = \nabla^2_{\theta_{b(K)}} \mathcal{L}(\theta_b; \epsilon_b) = [\mathbf{h}^1_b, \mathbf{h}^2_b, \cdots, \mathbf{h}^D_b]$ denote the Hessian at $\theta_b$ calculated on $\epsilon_b$, where $\mathbf{h}^d_b$ is the d-th column vector of $\mathbf{H}^j_{M_b}$. Similarly, $\widehat{\mathbf{H}}^j_{M_{b-1}} = [\hat{\mathbf{h}}^1_{b-1}, \cdots, \hat{\mathbf{h}}^D_{b-1}]$ denotes the Hessian for the momentum term at the $(b-1)$-th iteration. If $(\sum^D_{d=1} \lambda_d \mathbf{h}^d_b)(\sum^D_{d=1} \lambda_d \hat{\mathbf{h}}^d_{b-1}) \geq 0$ holds for any $\lambda_d \neq 0$, $d = \{1, 2, \cdots, D\}$, then we have the following iterative update rule for VR-MCL,*

$$\theta :\approx \theta - \alpha \hat{\mathbf{g}}^{\epsilon_b}_{\theta_b} = \theta - \alpha (\mathbf{H}^j_{\mathrm{VR}})^{-1} \nabla_\theta \mathcal{L}^j(\theta),$$

*where $(\mathbf{H}^j_{\mathrm{VR}})^{-1} = (\mathbf{H}^j_{M_b})^{-1} + r((\widehat{\mathbf{H}}^j_{M_{b-1}})^{-1} - (\overline{\mathbf{H}}^j_{M_{b-1}})^{-1})$ with $\overline{\mathbf{H}}^j_{M_{b-1}}$ denoting the Hessian at $\theta_{b-1}$ calculated on $\epsilon_b$. The eigenvalues of $(\mathbf{H}^j_{\mathrm{VR}})^{-1}$ are approximately given by $v^{-1}_b + r((\hat{v}_{b-1})^{-1} - (\overline{v}_{b-1})^{-1})$, where $v_b$, $\hat{v}_{b-1}$, and $\overline{v}_{b-1}$ denote the eigenvalues of $\mathbf{H}^j_{M_b}$, $\widehat{\mathbf{H}}^j_{M_{b-1}}$, and $\overline{\mathbf{H}}^j_{M_{b-1}}$, respectively.*

Please kindly refer to Appendix A.3 for the detailed proof. Proposition 3 develops the relationship between the Hessian $\mathbf{H}^j_{\mathrm{VR}}$ used in the update rule of VR-MCL and that used in the update rule of Meta-CL (i.e., $\mathbf{H}^j_{M_b}$). Similar to the analysis of the effect of Hessian in Secs. 2 and 3, we reach the following conclusions.

○ For those wrongly estimated low-curvature directions by Meta-CL, i.e., $\hat{v}_{b-1} \gg v_b, \overline{v}_{b-1}$ (some potential crucial weights are erroneously viewed as unimportant), we have $v^{-1}_b + r((\hat{v}_{b-1})^{-1} - (\overline{v}_{b-1})^{-1}) \approx r(\hat{v}_{b-1})^{-1} + (1-r)v^{-1}_b \leq v^{-1}_b$. This signifies that VR-MCL can prevent excessive updates triggered by the wrongly estimated low-curvature direction of the Hessian, thereby retaining the critical weights for previous tasks and improving the accuracy of its iterative updating formula.

○ For the high-curvature directions with large eigenvalues, i.e., $\hat{v}_{b-1} \approx v_b \approx \overline{v}_{b-1}$, we have $v^{-1}_b + r((\hat{v}_{b-1})^{-1} - (\overline{v}_{b-1})^{-1}) \approx v^{-1}_b$. This suggests that VR-MCL can effectively ensure cautious updates along these high-curvature directions of Meta-CL.

To sum up, Meta-CL is susceptible to inaccurate estimations of $\mathbf{H}^j_{M_b}$ due to the random samples drawn from $\mathcal{M}$. In contrast, the proposed VR-MCL incorporates a regularization term on $\mathbf{H}^j_{M_b}$, which effectively limits the model from making large updates and improves the precision of the iterative update. As a result, VR-MCL can effectively preserve the important parameters for previous tasks, thereby maintaining their performance.

**Theorem 1** (Regret Bound of VR-MCL). *The regret in online continual learning follows $\mathbf{CR}_j = \tilde{F}(\theta) - F(\theta^*)$, where $\tilde{F}(\theta) = \sum^j_{i=1} \mathcal{L}^i(\hat{\theta}^j)$, $F(\theta^*) = \min_\theta \sum^j_{i=1} \mathcal{L}^i(\theta)$. Assuming F with $\varphi$-Lipschitz hessian is $\mu$-strongly convex, G-Lipschitz, $\eta$-smooth and grounded on the four assumptions in Appendix A.4, $\sigma$ and M are two large constants in Assumption 3 and 4 respectively, and T denotes the training iteration. Then with probability at least $1 - \delta$ for any $\delta \in (0, 1)$, we can get,*

$$\mathbf{CR}_j \leq (\log T + 1)(F(\theta_1) - F(\theta^*)) + \frac{LD^2(\log T + 1)^2}{2}$$
$$+ \frac{LD^2(\log T + 1)^2}{2} + (16LD^2 + 16\sigma D + 4M)\sqrt{2T \log(8T/\delta)} = \tilde{\mathcal{O}}(\sqrt{T}).$$

Theorem 1 illustrates that, under mild assumptions in a stochastic setting, VR-MCL attains a nearly optimal $\tilde{\mathcal{O}}(\sqrt{T})$ regret bound w.h.p. for the online optimization problem. This theoretically demonstrates the effectiveness of the proposed VR-MCL algorithm. Detailed proof see Appendix A.4.

## 5 EXPERIMENTS

**Datasets and Settings.** To verify the effectiveness of the proposed VR-MCL, we conduct comprehensive experiments on commonly used datasets Seq-CIFAR10, as well as the longer task sequences

Table 2: Performance of Seq-CIFAR10 and longer task sequences Seq-CIFAR100, Seq-TinyImageNet with 95% confidence interval on reduced ResNet-18. The memory buffer size is set as 1000 (i.e., $|\mathcal{M}|$=1000). All reported numbers are the average of 5 runs. Shaded areas are our methods, and '–' indicates the result is omitted due to high instability.

| Method | Seq-CIFAR10 | | Seq-CIFAR100 | | Seq-TinyImageNet | |
|---|---|---|---|---|---|---|
| | AAA | Acc | AAA | Acc | AAA | Acc |
| SGD | $34.85_{\pm1.71}$ | $16.96_{\pm0.60}$ | $11.63_{\pm0.38}$ | $5.27_{\pm0.28}$ | $8.99_{\pm0.39}$ | $3.86_{\pm0.20}$ |
| LWF | $35.31_{\pm0.98}$ | $18.84_{\pm0.10}$ | $11.98_{\pm0.40}$ | $5.63_{\pm0.37}$ | $9.21_{\pm0.37}$ | $4.01_{\pm0.29}$ |
| A-GEM | $38.66_{\pm0.79}$ | $18.46_{\pm0.17}$ | $13.15_{\pm0.23}$ | $6.02_{\pm0.17}$ | $9.81_{\pm0.31}$ | $4.07_{\pm0.18}$ |
| GEM | $38.67_{\pm0.77}$ | $18.49_{\pm0.15}$ | $15.18_{\pm0.38}$ | $8.30_{\pm0.58}$ | $10.99_{\pm0.32}$ | $5.15_{\pm0.31}$ |
| ER | $55.53_{\pm2.58}$ | $43.83_{\pm4.84}$ | $23.19_{\pm0.38}$ | $16.07_{\pm0.88}$ | $19.45_{\pm0.40}$ | $11.13_{\pm0.39}$ |
| DER | $45.85_{\pm1.62}$ | $29.87_{\pm2.95}$ | $13.35_{\pm0.36}$ | $6.12_{\pm0.18}$ | $10.35_{\pm0.33}$ | $4.08_{\pm0.13}$ |
| DER++ | $64.22_{\pm0.70}$ | $52.29_{\pm1.86}$ | $19.88_{\pm0.43}$ | $11.79_{\pm0.65}$ | $14.75_{\pm0.15}$ | $8.26_{\pm0.34}$ |
| CLSER | $63.02_{\pm1.54}$ | $52.80_{\pm1.66}$ | $25.46_{\pm0.57}$ | $17.88_{\pm0.69}$ | $19.43_{\pm0.37}$ | $11.09_{\pm0.24}$ |
| OCM | $66.14_{\pm0.95}$ | $53.39_{\pm1.00}$ | $22.54_{\pm0.79}$ | $14.40_{\pm0.82}$ | $10.45_{\pm0.65}$ | $4.53_{\pm0.54}$ |
| ER-OBC | $65.82_{\pm0.91}$ | $54.85_{\pm2.16}$ | $25.54_{\pm0.25}$ | $17.21_{\pm0.92}$ | $20.11_{\pm0.31}$ | $11.51_{\pm0.18}$ |
| On-EWC | $38.44_{\pm0.50}$ | $17.12_{\pm0.51}$ | $11.81_{\pm0.42}$ | $5.88_{\pm0.31}$ | $9.38_{\pm0.17}$ | $3.65_{\pm0.22}$ |
| IS | $37.33_{\pm0.23}$ | $17.39_{\pm0.19}$ | $12.32_{\pm0.22}$ | $5.20_{\pm0.18}$ | $8.73_{\pm0.25}$ | $3.33_{\pm0.28}$ |
| MER | $50.99_{\pm0.65}$ | $36.92_{\pm2.42}$ | – | – | – | – |
| La-MAML | $42.98_{\pm1.60}$ | $33.43_{\pm1.21}$ | $12.55_{\pm0.39}$ | $11.78_{\pm0.65}$ | $11.10_{\pm0.70}$ | $6.74_{\pm0.36}$ |
| VR-MCL | $\mathbf{66.97}_{\pm\mathbf{1.58}}$ | $\mathbf{56.48}_{\pm\mathbf{1.79}}$ | $\mathbf{27.01}_{\pm\mathbf{0.48}}$ | $\mathbf{19.49}_{\pm\mathbf{0.69}}$ | $\mathbf{21.26}_{\pm\mathbf{0.53}}$ | $\mathbf{13.27}_{\pm\mathbf{0.39}}$ |

Seq-CIFAR100 and Seq-TinyImageNet (Buzzega et al., 2020). Specifically, the Seq-CIFAR10 dataset comprises 5 tasks, with each task containing 2 classes. In contrast, Seq-CIFAR100 consists of 10 tasks, each with 10 classes, while Seq-TinyImageNet includes 20 tasks, each encompassing 10 classes. In this paper, we focus on the following settings of continual learning:

- **Online CL:** the data is fed to the model in small batches and trained in a single pass.
- **Class Incremental:** the model cannot get the oracle task index during testing.

Our evaluation includes the metrics and experimental settings following the previous works on *online* CL with a single-head (Caccia et al., 2021; Ji et al., 2020; Shim et al., 2021). We choose the final *Averaged accuracy (Acc)* across all tasks after sequential training on each task as the main metric for comparing approaches. Moreover, under online CL, we use the *Averaged Anytime Accuracy (AAA)* Caccia et al. (2021) to evaluate the model through the stream tasks. Let $AA_j$ denote the test average accuracy after training on $\mathcal{T}_j$, then the evaluation metrics of *AAA* and *Acc* are:

$$AAA = \frac{1}{N}\sum_{j=1}^{N}(AA_j), \ Acc = AA_N.$$

**Baselines and Training Details.** To more effectively validate the efficacy of VR-MCL, which improves the Hessian component and thus better utilizes second-order information, we select key regularization-based CL methods (Huszár, 2017; Zenke et al., 2017) and Meta-CL methods (Riemer et al., 2019; Gupta et al., 2020). We also opt for other representative SOTA CL methods (Chaudhry et al., 2018; Lopez-Paz & Ranzato, 2017; Rolnick et al., 2019; Buzzega et al., 2020; Arani et al., 2021; Guo et al., 2022; Chrysakis & Moens, 2023) as baselines to further highlight the superior performance of VR-MCL. For detailed information on each baseline method, please refer to Appendix C. For fair comparison among different CL methods, we train all the models using the Stochastic Gradient Descent (SGD) optimizer under the online continual learning. Following (Chrysakis & Moens, 2023; Lopez-Paz & Ranzato, 2017) and (Gupta et al., 2020) we utilize two backbones: reduced ResNet-18 and a smaller network named PcCNN which has only 3 conv-layers. For the reduced ResNet18, we set both the batch size and the replay batch size (i.e., the batch size sampled from the memory buffer $\mathcal{M}$) as 32. For the smaller network PcCNN, we set both the batch size and the replay bath size as 10. The momentum ratio $r$ and the learning rate $\alpha$ of VR-MCL are both set as 0.25 for all experiments. For other training details please check Appendix C.

To assess the effectiveness of VR-MCL, we perform a comprehensive set of experiments and conduct ablation studies to address the following six key questions:

**Question 1: How does VR-MCL perform on online CL benchmarks?** We report results with a 95% confidence interval on Seq-CIFAR10 and the longer task sequences Seq-CIFAR100, Seq-TinyImageNet in Table 2. The methods in the bottom row are the methods utilizing second-order information while the top row are the methods of other representative CL methods. It is evident that the proposed VR-MCL substantially enhances the performance of CL methods employing second-

Table 3: Performance of Seq-CIFAR100 with 95% confidence interval (CI) on different memory buffer sizes $|\mathcal{M}|$. The backbone is reduced ResNet-18, and all numbers are the average of 5 runs. Shaded areas are our methods, and '–' indicates the result is omitted due to high instability.

| Method | $|\mathcal{M}|$=200 AAA | $|\mathcal{M}|$=200 Acc | $|\mathcal{M}|$=600 AAA | $|\mathcal{M}|$=600 Acc | $|\mathcal{M}|$=1000 AAA | $|\mathcal{M}|$=1000 Acc |
|---|---|---|---|---|---|---|
| SGD | $11.63_{\pm0.38}$ | $5.60_{\pm0.28}$ | $11.63_{\pm0.38}$ | $5.60_{\pm0.28}$ | $11.63_{\pm0.38}$ | $5.60_{\pm0.28}$ |
| LWF | $12.01_{\pm0.42}$ | $5.84_{\pm0.21}$ | $12.12_{\pm0.41}$ | $5.97_{\pm0.33}$ | $12.01_{\pm0.43}$ | $5.77_{\pm0.32}$ |
| A-GEM | $13.45_{\pm0.29}$ | $6.25_{\pm0.25}$ | $13.36_{\pm0.21}$ | $6.00_{\pm0.16}$ | $13.15_{\pm0.36}$ | $6.12_{\pm0.18}$ |
| GEM | $15.41_{\pm0.21}$ | $7.92_{\pm0.35}$ | $14.92_{\pm0.24}$ | $7.89_{\pm0.31}$ | $15.18_{\pm0.38}$ | $8.30_{\pm0.58}$ |
| ER | $18.52_{\pm0.63}$ | $10.15_{\pm0.42}$ | $21.73_{\pm0.25}$ | $13.63_{\pm0.60}$ | $23.19_{\pm0.38}$ | $16.07_{\pm0.88}$ |
| DER | $13.44_{\pm0.31}$ | $6.12_{\pm0.12}$ | $13.57_{\pm0.27}$ | $6.23_{\pm0.12}$ | $13.35_{\pm0.36}$ | $6.12_{\pm0.18}$ |
| DER++ | $17.91_{\pm0.47}$ | $9.59_{\pm0.42}$ | $19.13_{\pm0.27}$ | $10.45_{\pm0.30}$ | $19.88_{\pm0.43}$ | $11.79_{\pm0.65}$ |
| CLSER | $20.56_{\pm0.35}$ | $11.35_{\pm0.36}$ | $23.63_{\pm0.33}$ | $16.55_{\pm0.35}$ | $25.46_{\pm0.57}$ | $17.88_{\pm0.69}$ |
| OCM | – | – | $18.73_{\pm0.10}$ | $8.40_{\pm0.37}$ | $22.54_{\pm0.79}$ | $14.40_{\pm0.82}$ |
| ER-OBC | $19.53_{\pm0.28}$ | $10.38_{\pm0.37}$ | $23.96_{\pm0.40}$ | $15.83_{\pm0.41}$ | $25.54_{\pm0.25}$ | $17.21_{\pm0.92}$ |
| On-EWC | $11.81_{\pm0.42}$ | $5.88_{\pm0.31}$ | $11.81_{\pm0.42}$ | $5.88_{\pm0.31}$ | $11.81_{\pm0.42}$ | $5.88_{\pm0.31}$ |
| IS | $12.32_{\pm0.22}$ | $5.20_{\pm0.18}$ | $12.32_{\pm0.22}$ | $5.20_{\pm0.18}$ | $12.32_{\pm0.22}$ | $5.20_{\pm0.18}$ |
| La-MAML | $8.48_{\pm0.34}$ | $6.95_{\pm0.27}$ | $10.07_{\pm1.10}$ | $9.05_{\pm0.58}$ | $12.55_{\pm0.39}$ | $11.78_{\pm0.65}$ |
| VR-MCL | $\mathbf{22.41_{\pm0.59}}$ | $\mathbf{13.27_{\pm0.40}}$ | $\mathbf{25.97_{\pm0.47}}$ | $\mathbf{17.15_{\pm0.59}}$ | $\mathbf{27.01_{\pm0.48}}$ | $\mathbf{19.49_{\pm0.69}}$ |

order information, and also outperforms other representative SOTA CL methods by nearly 2%. It is noteworthy that the performance of VR-MCL is superior to both MER (Riemer et al., 2019) and La-MAML (Gupta et al., 2020), and significantly better than IS (Zenke et al., 2017) and On-EWC (Huszár, 2017). This observation aligns with our analysis in Table 1.

**Question 2: How does the performance gain from VR-MCL vary with buffer size $|\mathcal{M}|$?** In large-scale CL, it is impractical to store a large number of examples from previous tasks due to storage constraints. Therefore, an ideal CL method should still perform well even with a modest memory buffer. To investigate if VR-MCL is still effective with a smaller buffer size, we conduct experiments on Seq-CIFAR100 with different $|\mathcal{M}|$ as shown in Table 3. The results show that the performance of most methods improves with the increase of memory buffer size $|\mathcal{M}|$. And our VR-MCL consistently outperforms other baselines, suggesting that VR-MCL is effective even with modest buffer sizes. Note that GEM (Lopez-Paz & Ranzato, 2017) and A-GEM (Chaudhry et al., 2018) are insensitive to the buffer size complying with the results in Shim et al. (2021).

**Question 3: How effective is VR-MCL in dealing with increasingly non-stationary Settings (i.e., imbalanced CL)?** In the imbalanced CL setting, where the number of samples varies across tasks, an imbalance issue arises in the samples stored in $\mathcal{M}$. This introduces a high variance during the calculation process of the hyper-gradient, thereby negatively affecting the impact on the implicit Hessian. To assess the effectiveness of VR-MCL in addressing these challenges, we choose different imbalanced settings and explore the performance of various methods under this setup in Table 4. Specifically, we choose three imbalanced settings. The *Normal* means the total number of samples per streaming task from high to low, and the *Reversed* is the opposite. The *Random* setting, on the other hand, represents a situation where there is no specific pattern in the number of samples per streaming task. From Table 4, we can see that the performance of most models degrades in this challenging setup. However, VR-MCL significantly outperforms other baselines and even surpasses methods such as CBRS (Chrysakis & Moens, 2020) and Coresets (Borsos et al., 2020), which are specifically designed for imbalanced CL. This suggests that the VR-MCL has excellent performance under the challenged imbalanced CL setting. The 95% CI for Table 4, as well as additional experimental results for different datasets and various imbalance ratios, can be found in Appendix D.4.

**Question 4: Whether the proposed method VR-MCL is still effective using different backbones?** To verify whether the improvement of VR-MCL is related to the backbone network, we change the reduced ResNet-18 as a shallow network PcCNN following Gupta et al. (2020). The outcomes are provided in Table 5. It can be observed that VR-MCL shows a nearly 3% improvement over the other best method in terms of both the $AAA$ and $Acc$ metrics. This indicates that the proposed VR-MCL remains effective across different backbone architectures.

**Question 5: How does VR-MCL compare to other variance reduction methods?** As discussed in Sec. 4.1, most existing popular variance reduction methods like SAG (Gower et al., 2020), SAGA (Defazio et al., 2014), and SVRG (Johnson & Zhang, 2013) cannot be directly applied since they need to compute the full-batch gradient which is not accessible in online CL. Therefore, we

compare VR-MCL with other techniques that promise to reduce the variance: 1) VR-MCL[1], which simply increases the **replay batch size**; 2) VR-MCL[2], which incorporates SGD with the naive momentum term. The outcomes shown in Table 6 indicate that: 1) Other variance reduction techniques are also effective, supporting our analysis in Sec. 4; 2) Our proposed momentum-based VR-MCL significantly outperforms other variance techniques, further demonstrating its effectiveness.

Table 4: Performance on the imbalanced Seq-CIFAR10 ($|\mathcal{M}|$=1000) with imbalance ratio $\gamma$=2. The *Reversed* means the total number of samples per streaming task from low to high. Full table see Appendix D.6.

| Methods | $\gamma$ =2 (*Normal*) AAA | Acc | $\gamma$ =2 (*Reversed*) AAA | Acc | $\gamma$ =2 (*Random*) AAA | Acc |
|---|---|---|---|---|---|---|
| SGD | 36.64 | 16.46 | 35.57 | 17.54 | 34.62 | 17.33 |
| A-GEM | 38.50 | 17.64 | 36.72 | 17.43 | 37.79 | 17.85 |
| GEM | 41.29 | 17.90 | 38.46 | 18.37 | 39.63 | 18.00 |
| ER | 52.50 | 33.10 | 46.58 | 28.49 | 43.78 | 31.10 |
| DER | 46.95 | 16.82 | 40.62 | 18.63 | 41.44 | 18.78 |
| DER++ | 62.02 | 44.14 | 58.22 | 39.21 | 60.25 | 42.83 |
| CLS-ER | 61.37 | 47.75 | 54.92 | 40.51 | 56.60 | 47.48 |
| On-EWC | 38.85 | 16.92 | 37.41 | 17.79 | 37.26 | 14.35 |
| CBRS | 59.07 | 43.81 | 57.57 | 44.20 | 58.16 | 44.66 |
| Coresets | 61.11 | 45.37 | 58.12 | 45.80 | 58.56 | 45.63 |
| La-MAML | 36.64 | 29.17 | 32.08 | 31.17 | 42.52 | 31.24 |
| VR-MCL | **65.06** | **49.82** | **61.16** | **51.36** | **61.91** | **50.74** |

Table 5: Performance of Seq-CIFAR10 with 95% confidence interval on the small network PcCNN with $|\mathcal{M}|$=200. The results are the average of 5 runs.

| Method | AAA | Acc |
|---|---|---|
| SGD | $37.84_{\pm 0.03}$ | $17.58_{\pm 0.20}$ |
| A-GEM | $41.84_{\pm 0.44}$ | $18.50_{\pm 0.38}$ |
| GEM | $45.07_{\pm 0.52}$ | $22.52_{\pm 1.02}$ |
| ER | $54.64_{\pm 0.91}$ | $32.46_{\pm 1.58}$ |
| DER | $58.06_{\pm 0.18}$ | $38.21_{\pm 0.29}$ |
| DER++ | $56.07_{\pm 0.32}$ | $34.28_{\pm 0.79}$ |
| CLS-ER | $58.03_{\pm 0.75}$ | $39.38_{\pm 1.44}$ |
| ER-OBC | $56.75_{\pm 0.68}$ | $38.10_{\pm 1.28}$ |
| On-EWC | $38.72_{\pm 0.58}$ | $17.05_{\pm 0.08}$ |
| MER | $55.50_{\pm 0.38}$ | $32.01_{\pm 1.22}$ |
| La-MAML | $46.36_{\pm 0.98}$ | $29.45_{\pm 0.51}$ |
| VR-MCL | $\mathbf{60.88_{\pm 0.22}}$ | $\mathbf{43.41_{\pm 0.30}}$ |

Table 6: Performance with 95% CI using various variance reduction techniques, where $|\mathcal{M}| = 1000$.

| Method | Seq-CIFAR10 AAA | Acc | Seq-CIFAR100 AAA | Acc | Seq-TinyImageNet AAA | Acc |
|---|---|---|---|---|---|---|
| La-MAML | $42.98_{\pm 1.60}$ | $33.43_{\pm 1.21}$ | $12.55_{\pm 0.39}$ | $11.78_{\pm 0.65}$ | $11.10_{\pm 0.70}$ | $6.74_{\pm 0.36}$ |
| VR-MCL[1] | $66.74_{\pm 2.49}$ | $55.02_{\pm 3.70}$ | $26.15_{\pm 0.53}$ | $19.02_{\pm 0.45}$ | $20.39_{\pm 1.17}$ | $12.09_{\pm 1.30}$ |
| VR-MCL[2] | $62.87_{\pm 1.70}$ | $53.20_{\pm 1.52}$ | $20.15_{\pm 1.52}$ | $16.05_{\pm 2.04}$ | $15.96_{\pm 0.51}$ | $9.82_{\pm 0.45}$ |
| VR-MCL(ours) | $\mathbf{66.97_{\pm 1.58}}$ | $\mathbf{56.48_{\pm 1.79}}$ | $\mathbf{27.01_{\pm 0.48}}$ | $\mathbf{19.49_{\pm 0.69}}$ | $\mathbf{21.26_{\pm 0.53}}$ | $\mathbf{13.27_{\pm 0.39}}$ |

**Question 6: Whether the proposed method VR-MCL can effectively reduce the variance of gradients?** To explore this problem, following (Yang & Kwok, 2022), we choose the relative variance metric (i.e., $\frac{E\|\mathbf{g}_{\theta_t} - E(\mathbf{g}_{\theta_t})\|^2}{\|E(\mathbf{g}_{\theta_t})\|^2}$), which can eliminate the influence of the gradient value on the variance. The outcomes are shown in Fig. 3, where the Meta-CL refers to La-MAML (Gupta et al., 2020). It can be seen that the proposed VR-MCL indeed has less variance during training, complying with the analysis in Eqn. (5).

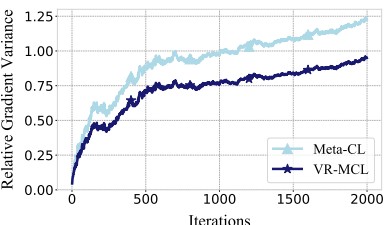

Figure 3: The relative variance of Meta-CL and the proposed VR-MCL.

**Other Results.** Due to space constraints, additional comprehensive experiments have been provided in Appendix D, including other evaluation metrics (i.e., forgetting measure), hyperparameter selection (i.e., the selection of the momentum ratio $r$ and learning rate $\alpha$), and training time analysis.

## 6 CONCLUSION

In this paper, we revisited Meta-CL and first bridged it with regularization-based methods from the Hessian matrix approximation perspective. Specifically, Meta-CL, through the use of hypergradient, implicitly approximates the Hessian in an online manner. While this approach benefits from timely adaptation, it also grapples with high variance resulting from random memory buffer sampling. Building on this understanding, we proposed the VR-MCL method, which effectively reduces the variance of the hypergradient and exhibits superior performance under the online continual learning setting. We provide theoretical proof that VR-MCL imposes a regularization term on the implicitly estimated Hessian matrix. This prevents updates from moving excessively in the wrongly estimated low-curvature directions and thus has a more accurate iterative update rule shown in Table 1. Moreover, to enhance comprehension, we also provide the regret bound of our VR-MCL.

## ACKNOWLEDGEMENT

We thank all the anonymous reviewers for their constructive suggestions on improving this paper. This research was sponsored by the National Key R&D Program of China (2020YFA0713900), the RMGS 9229073, the China NSFC projects under contract 62272375, 12226004 and 62306233, the Young Elite Scientists Sponsorship Program by CAST 2023QNRC001.

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

# A    DETAILED PROOF

## A.1    PROOF OF PROPOSITION 1

**Proposition 1.** *In regularization-based continual learning, if the model parameter $\theta$ is searched within the neighborhood set $\cup_{i=1}^{j-1}\mathcal{N}^i$ with $\mathcal{N}^i = \{\theta : d(\theta, \hat{\theta}^i) < \delta^i\}$, then the iterative update rule of $\theta$ approximately is*

$$\theta :\approx \theta - \alpha(\mathbf{H}^1 + \mathbf{H}^2 + \cdots + \mathbf{H}^{j-1})^{-1}\nabla_\theta \mathcal{L}^j(\theta)$$

*Proof.* The empirical loss $\mathcal{L}$ on $\mathcal{T}^{[1:j]}$ can be approximate as,

$$
\begin{aligned}
\mathcal{L} &= \sum_{i=1}^{j-1}\mathcal{L}_i(\theta) + \mathcal{L}_j(\theta) \approx \mathcal{L}_{j-1}^{\mathrm{prox}}(\theta) + \mathcal{L}_j(\theta) \\
&\stackrel{(1)}{=} \sum_{i=1}^{j-1} \underbrace{\mathcal{L}^i(\hat{\theta}^i)}_{(a)} + \underbrace{(\theta - \hat{\theta}^i)^\top \nabla_\theta \mathcal{L}^i(\hat{\theta}^i)}_{(b)} + \underbrace{\frac{1}{2}(\theta - \hat{\theta}^i)^\top \mathbf{H}^i (\theta - \hat{\theta}^i)}_{(c)} + \mathcal{L}_j(\theta) \\
&\stackrel{(2)}{\approx} \sum_{i=1}^{j-1}\frac{1}{2}(\theta - \hat{\theta}^i)^\top \mathbf{H}^i(\theta - \hat{\theta}^i) + \mathcal{L}_j(\theta)
\end{aligned}
$$

Here, (1) is expand each loss $\mathcal{L}^i$ using Taylor series at point $\hat{\theta}^i$ to compute $\mathcal{L}_{j-1}^{\mathrm{prox}}(\theta)$, where $i = 1, ..., j-1$. Step (2) is due to the term (a) is not related to $\theta$ and thus can be discarded. Furthermore, since the training process at the end of each task (i.e., $\hat{\theta}^i$) is usually converged, the gradient $\nabla_\theta \mathcal{L}^i(\hat{\theta}^i)$ is near zero and the term (b) can be ignored in practice. Then we can derive $\frac{\partial \mathcal{L}}{\partial \theta}$ as,

$$\frac{\partial \mathcal{L}}{\partial \theta} = \sum_{i=1}^{j-1}\mathbf{H}^i(\theta - \hat{\theta}^i) + \nabla_\theta \mathcal{L}_j(\theta).$$

If the parameters $\theta$ are searched in the neighborhood set $\cup_{i=1}^{j-1}\mathcal{N}^i$, where $\mathcal{N}^i = \{\theta : d(\theta, \hat{\theta}^i) < \delta^i\}$, then we can make the approximation $\sum_{i=1}^{j-1}\mathbf{H}^i(\theta - \hat{\theta}^i) \sim (\sum_{i=1}^{j-1}\mathbf{H}^i)(\theta - \hat{\theta}^{j-1})$ Huszár (2017). Let $\frac{\partial \mathcal{L}}{\partial \theta} = 0$, we can obtain the iterative update formula of $\mathcal{T}^j$ as:

$$\theta :\approx \theta - \alpha(\mathbf{H}^1 + \mathbf{H}^2 + \cdots + \mathbf{H}^{j-1})^{-1}\nabla_\theta \mathcal{L}^j(\theta),$$

where $\alpha = 1$ to maintain formal consistency with other iterative rules in Table 1, $(\mathbf{H}^1 + \mathbf{H}^2 + \cdots + \mathbf{H}^{j-1})$ are the Hessian matrices of previous tasks computed at the end of training of each $\mathcal{T}^i$, and $\nabla_\theta \mathcal{L}^j(\theta)$ is the gradient of the $j$-th task. $\qquad\square$

## A.2    PROOF OF PROPOSITION 2

**Proposition 2.** *For MCL with single inner step adaption, suppose that $\theta_{(K)}$ is located in the $\epsilon$-neighborhood $\mathcal{N}(\theta^*, \epsilon)$ of the optimal model parameter $\theta^* = \mathrm{argmin}_\theta \mathcal{L}^{[1:j]}(\theta_{(K)})$, $\mathcal{L}$ is $\mu$-smooth, and $\beta < \sqrt{\delta/|\nabla_\theta \mathcal{L}^j(\theta) - (\nabla_\theta \mathcal{L}^j(\theta))^2|}$ where $\delta$ is a small number. Thus, the iterative update rule approximately is*

$$\theta :\approx \theta - \alpha(\mathbf{H}_M^j)^{-1}\nabla_\theta \mathcal{L}^j(\theta),$$

*Proof.* **(Single inner step adaption)** For simplicity, we analyze the MCL with single inner step adaption (i.e., $K = 1$) at first. The objective function of one step inner-loop MCL could be formulated as,

$$\theta^* = \mathrm{argmin}_\theta \mathcal{L}^{[1:j]}(\theta_{(K)}), \qquad s.t. \ \theta_{(K)} = \theta - \beta\nabla_\theta \mathcal{L}^j(\theta).$$

Then the derivative of loss $\mathcal{L}^{[1:j]}(\theta_{(K)})$ over $\theta$ is,

$$\frac{\partial \mathcal{L}^{[1:j]}(\theta_{(K)})}{\partial \theta} = \frac{\partial \mathcal{L}^{[1:j]}(\theta_{(K)})}{\partial \theta_{(K)}}\frac{\partial \theta_{(K)}}{\theta}. \tag{1}$$

For the first term $\frac{\partial \mathcal{L}^{[1:j]}(\theta_{(K)})}{\partial \theta_{(K)}}$, we take the Taylor expansion at $\theta$,

$$\frac{\partial \mathcal{L}^{[1:j]}(\theta_{(K)})}{\partial \theta_{(K)}} = \nabla_{\theta_{(K)}} \mathcal{L}^{[1:j]}(\theta_{(K)}) \approx \nabla_{\theta_{(K)}} \mathcal{L}^{[1:j]}(\theta) + \mathbf{H}_M^j(\theta_{(K)} - \theta) + (\theta_{(K)} - \theta)^T \otimes \mathbf{T} \otimes (\theta_{(K)} - \theta),$$

(2)

where $\mathbf{H}_M^j = \nabla_{\theta_{(K)}}^2 \mathcal{L}^{[1:j]}(\theta)$ and $\mathbf{T}$ denote the Hessian matrix and the third-order symmetric tensor, respectively, and $\otimes$ represents the Kronecker product.

Since we hope to find the optimal parameters $\theta^*$ so as to achieve the lowest $\mathcal{L}^{[1:j]}$, which means $\frac{\mathcal{L}^{[1:j]}(\theta_{(K)})}{\partial \theta} = 0$. According to Eqn. (1), that is approximately to let $\frac{\partial \mathcal{L}^{[1:j]}(\theta_{(K)})}{\partial \theta_{(K)}} = 0$.

Suppose $\delta$ is a small number, $\beta < \sqrt{\delta/|\nabla_\theta \mathcal{L}^j(\theta) - (\nabla_\theta \mathcal{L}^j(\theta))^2|}$, and there exists an $\epsilon$-neighborhood $\mathcal{N}(\cdot)$ such that parameters $\theta^* \in \mathcal{N}(\theta - \beta \nabla_\theta \mathcal{L}^j(\theta), \epsilon)$. Since $\mathcal{L}$ is $\mu$-smooth, which means $\|\nabla \mathcal{L}(\theta) - \nabla \mathcal{L}(\theta')\|_2 \leq \mu \|\theta - \theta'\|_2$. Then we can get the following equations according to Eqn. (2),

$$\begin{aligned}
\frac{\partial \mathcal{L}^{[1:j]}(\theta^*)}{\partial \theta^*} &= \nabla_{\theta^*} \mathcal{L}^{[1:j]}(\theta^*) \\
&\approx \nabla_{\theta^*} \mathcal{L}^{[1:j]}(\theta) + \nabla_{\theta^*}^2 \mathcal{L}^{[1:j]}(\theta)(\theta^* - \theta) + (\theta^* - \theta)^T \otimes T \otimes (\theta^* - \theta) \\
&\approx \nabla_{\theta_{(K)}} \mathcal{L}^{[1:j]}(\theta) + \nabla_{\theta_{(K)}}^2 \mathcal{L}^{[1:j]}(\theta)(\theta^* - \theta) + (\theta_{(K)} - \theta)^T \otimes T \otimes (\theta_{(K)} - \theta) + o(\epsilon) \\
&= \nabla_{\theta_{(K)}} \mathcal{L}^{[1:j]}(\theta) + \nabla_{\theta_{(K)}}^2 \mathcal{L}^{[1:j]}(\theta)(\theta^* - \theta) + C \odot (\theta_{(K)} - \theta)^T (\theta_{(K)} - \theta) + o(\epsilon) \\
&\approx o(\mu\epsilon) + \nabla_{\theta_{(K)}}^2 \mathcal{L}^{[1:j]}(\theta)(\theta^* - \theta) + \beta^2 (\nabla_\theta \mathcal{L}^j(\theta))^2 + o(\epsilon) \\
&\approx o(\mu\epsilon) + \nabla_{\theta_{(K)}}^2 \mathcal{L}^{[1:j]}(\theta)(\theta^* - \theta) + \beta^2 \nabla_\theta \mathcal{L}^j(\theta) + o(\delta) + o(\epsilon) \\
&\approx \nabla_{\theta_{(K)}}^2 \mathcal{L}^{[1:j]}(\theta)(\theta^* - \theta) + \beta^2 \nabla_\theta \mathcal{L}^j(\theta) + o(\delta) + o(\epsilon) + o(\mu\epsilon) = 0,
\end{aligned}$$

(3)

where $C$ means a constant vector, $\odot$ denotes the element-wise multiplication operator. Then it is easy to get $\theta := \theta - \beta^2 (\nabla_{\theta_{(K)}}^2 \mathcal{L}^{[1:j]}(\theta))^{-1} \nabla_\theta \mathcal{L}^j(\theta) = \theta - \alpha(\mathbf{H}_M^j)^{-1} \nabla_\theta \mathcal{L}^j(\theta)$, where $\alpha = \beta^2$ and $\mathbf{H}_M^j = \nabla_{\theta_{(K)}}^2 \mathcal{L}^{[1:j]}(\theta)$ .

**(K inner steps adaption)** For MCL with K steps of inner-loop updating, $\theta_{(K)} = \theta - \beta \nabla \mathcal{L}_{(1)}^j(\theta) - \beta \nabla \mathcal{L}_{(2)}^j(\theta_{(1)}) - ... - \beta \nabla \mathcal{L}_{(K)}^j(\theta_{(K-1)})$. Similar to Eqn. (3), we can get

$$\theta := \theta - \alpha(\mathbf{H}_M^j)^{-1} \Big( \sum_{i=0}^{K-1} \nabla \mathcal{L}_{(i+1)}^j(\theta_{(K-1)}) \Big),$$

where $\theta_{(i)}$ denotes the model parameters after the $i$-th inner loop adaptation and $\theta_{(0)} = \theta$. It is worth noting that in MCL with multiple updating steps, the hypergradient incorporates more gradients from the inner loop, which is the main difference from the one-step inner loop MCL. However, this difference does not affect the variance reduction analysis presented in Proposition 3. $\square$

## A.3 PROOF OF PROPOSITION 3

### A.3.1 PROOF OF LEMMA 1

**Lemma 1.** *(The linear combination of two invertible matrices is invertible under finite conditions.) Let $A = [a_1, a_2, ..., a_n] \in R^{n \times n}$ and $B = [b_1, b_2, ..., b_n] \in R^{n \times n}$ denote two square matrices, where $a_i \in R^n$, $b_i \in R^n$ are the $i$-th column vector of the matrix $A$ and $B$. Suppose: 1) $A$ and $B$ are invertible ; 2) for any $\lambda_i \neq 0, i = \{1, 2, ..n\}$, $(\sum_{i=1}^n \lambda_i a_i)(\sum_{i=1}^n \lambda_i b_i) \geq 0$, then the linear combination of $A$ and $B$ (i.e., $rA + (1-r)B$) is invertible, $r \in (0, 1)$.*

*Proof.* **(By contradiction.)**

**(Step-1)** We assume $rA + (1-r)B$, where $r \in (0, 1)$, is not invertible, which means the square matrix $rA + (1-r)B = [ra_1 + (1-r)b_1, ..., ra_n + (1-r)b_n]$ is singular.

**(Step-2)** Then there must exists $\lambda_i \neq 0, i = \{1, 2, ..n\}$, such that $\sum_{i=1}^{n} \lambda_i (ra_i + (1-r)b_i) = 0$. That is equivalent to $r \sum_{i=1}^{n} \lambda_i a_i + (1-r) \sum_{i=1}^{n} \lambda_i b_i = 0$. According to condition 2, it can deduce that $\sum_{i=1}^{n} \lambda_i a_i = \sum_{i=1}^{n} \lambda_i b_i = 0$. This is contradict with the condition that $A$ and $B$ are invertible.

**(Step-3)** Since assuming $rA + (1-r)B$ is not invertible leads to a contradiction, therefore, $rA + (1-r)B$ is convertible. $\qquad \square$

### A.3.2 PROOF OF PROPOSITION 3

**Proposition 3.** *Assume that the batch size for inner step adaptation is sufficiently large. Let $\mathbf{H}_{M_b}^{j} = \nabla_{\theta_{b(K)}}^{2} \mathcal{L}^{[1:j]_b}(\theta_b) = [\mathbf{h}_b^1, \mathbf{h}_b^2, \cdots, \mathbf{h}_b^D]$ denote the Hessian at $\theta_b$ calculated on $\epsilon_b$, where $\mathbf{h}_b^d$ is the d-th column vector of $\mathbf{H}_{M_b}^{j}$. Similarly, $\widehat{\mathbf{H}}_{M_{b-1}}^{j} = [\hat{\mathbf{h}}_{b-1}^1, \cdots, \hat{\mathbf{h}}_{b-1}^D]$ denotes the Hessian for the momentum term at the $(b-1)$-th iteration. If $(\sum_{d=1}^{D} \lambda_d \mathbf{h}_b^d)(\sum_{d=1}^{D} \lambda_d \hat{\mathbf{h}}_{b-1}^d) \geq 0$ holds for any $\lambda_d \neq 0, d = \{1, 2, \cdots, D\}$, then we have the following iterative update rule for VR-MCL,*

$$\theta :\approx \theta - \alpha(\mathbf{H}_{\text{VR}}^{j})^{-1} \nabla_\theta \mathcal{L}^j(\theta),$$

*where $(\mathbf{H}_{\text{VR}}^{j})^{-1} = (\mathbf{H}_{M_b}^{j})^{-1} + r((\widehat{\mathbf{H}}_{M_{b-1}}^{j})^{-1} - (\overline{\mathbf{H}}_{M_{b-1}}^{j})^{-1})$ with $\overline{\mathbf{H}}_{M_{b-1}}^{j}$ denoting the Hessian at $\theta_{b-1}$ calculated on $\epsilon_b$. The eigenvalues of $(\mathbf{H}_{\text{VR}}^{j})^{-1}$ are approximately given by $v_b^{-1} + r((\hat{v}_{b-1})^{-1} - (\overline{v}_{b-1})^{-1})$, where $v_b$, $\hat{v}_{b-1}$, and $\overline{v}_{b-1}$ denote the eigenvalues of $\mathbf{H}_{M_b}^{j}$, $\widehat{\mathbf{H}}_{M_{b-1}}^{j}$, and $\overline{\mathbf{H}}_{M_{b-1}}^{j}$, respectively.*

*Proof.* Let $\mathbf{g}_{\theta_b}^{\epsilon_b}$ denote the gradient of the $b$-th outer step. In proposition 2, the updating formula of MCL is $\theta := \theta - \alpha(\mathbf{H}_M^j)^{-1} \nabla_\theta \mathcal{L}^j(\theta)$ which means the gradient $\mathbf{g}_{\theta_b}^{\epsilon_b}$ can be rewrite in the form of $(\mathbf{H}_{M_b}^j)^{-1} \nabla_\theta \mathcal{L}^j(\theta)$. When we adopt the variance reduction method, let $m$ denote the momentum part, then we can get,

$$m_1 = \mathbf{g}_{\theta_1}^{\epsilon_1} = (\mathbf{H}_{M_1}^j)^{-1} \nabla_\theta \mathcal{L}^j(\theta_1),$$
$$m_2 = \mathbf{g}_{\theta_2}^{\epsilon_2} + r(m_1 - \mathbf{g}_{\theta_1}^{\epsilon_2}) = (\mathbf{H}_{M_2^j})^{-1} \nabla_\theta \mathcal{L}^j(\theta_2) + r((\mathbf{H}_{M_1}^j)^{-1} \nabla_\theta \mathcal{L}^j(\theta) - (\overline{\mathbf{H}}_{M_1}^j)^{-1} \nabla_\theta \overline{\mathcal{L}}^j(\theta_1)),$$
$$\cdots$$
$$m_b = \mathbf{g}_{\theta_b}^{\epsilon_b} + r(m_{b-1} - \mathbf{g}_{\theta_{b-1}}^{\epsilon_b}),$$

where $(\overline{\mathbf{H}}_{M_1}^j)^{-1}$ and $\nabla_\theta \overline{\mathcal{L}}^j(\theta)$ are the Hessian and corresponding gradient in $\mathbf{g}_{\theta_1}^{\epsilon_2}$. When there has a sufficiently large batch size on the current task, then $\nabla_\theta \mathcal{L}^j(\theta_2) \approx \nabla_\theta \overline{\mathcal{L}}^j(\theta_1) \approx \nabla_\theta \mathcal{L}^j(\theta_1)$. Therefore, we can rewrite $m_2$ as,

$$m_2 = \mathbf{g}_{\theta_2}^{\epsilon_2} + r(m_1 - \mathbf{g}_{\theta_1}^{\epsilon_2}) = \{(\mathbf{H}_{M_2}^j)^{-1} + r((\mathbf{H}_{M_1}^j)^{-1} - (\overline{\mathbf{H}}_{M_1})^{-1})\} \nabla_\theta \mathcal{L}^j(\theta_2).$$

Since $(\overline{\mathbf{H}}_{M_1}^j)^{-1}$ is highly related with $(\overline{\mathbf{H}}_{M_2}^j)^{-1}$, we can get,

$$(\mathbf{H}_{M_2}^j)^{-1} + r((\mathbf{H}_{M_1}^j)^{-1} - (\overline{\mathbf{H}}_{M_1}^j)^{-1}) \approx (1-r)(\mathbf{H}_{M_2}^j)^{-1} + r(\mathbf{H}_{M_1}^j)^{-1}$$

- For $m_2$, $\mathbf{H}_{M_1}^j = [h_1^1, h_1^2, ..., h_1^D]$ and $\mathbf{H}_{M_2}^j = [h_2^1, h_2^2, ..., h_2^D]$ are also highly related. Based on the assumption that for any $\lambda_d \neq 0, d = \{1, 2, ..D\}$, $(\sum_{d=1}^{D} \lambda_d h_1^d)(\sum_{d=1}^{D} \lambda_d h_2^d) \geq 0$. **According to Lemma 1**, we can get $r(\mathbf{H}_{M_1}^j)^{-1} + (1-r)(\mathbf{H}_{M_2}^j)^{-1}$ is invertible, and thus there exist a matrix $(\widehat{\mathbf{H}}_{M_2})^{-1} = r(\mathbf{H}_{M_1})^{-1} + (1-r)(\mathbf{H}_{M_2})^{-1}$.

- For $m_b$, the previous assumption also holds, which is for $\mathbf{H}_{M_b}^j = [h_b^1, ..., h_b^D]$, $\widehat{\mathbf{H}}_{M_{(b-1)}}^j = [\hat{h}_{b-1}^1, ..., \hat{h}_{b-1}^D]$, for any $\lambda_d \neq 0, d = \{1, 2, ..D\}$, $(\sum_{d=1}^{D} \lambda_d h_b^d)(\sum_{d=1}^{D} \lambda_d \hat{h}_{b-1}^d) \geq 0$. According to this recurrence relation, we can get $(\widehat{\mathbf{H}}_{M_b})^{-1} = r(\widehat{\mathbf{H}}_{M_{b-1}})^{-1} + (1-r)(\overline{\mathbf{H}}_{M_b})^{-1}$. Let $\mathbf{H}_{\text{VR}}^j = \widehat{\mathbf{H}}_{M_b}$, we can get the following iterative update rule of VR-MCL,

$$\theta :\approx \theta - \alpha(\mathbf{H}_{\text{VR}}^j)^{-1} \nabla_\theta \mathcal{L}^j(\theta),$$

$\qquad \square$

A.4    REGRET BOUND OF VR-MCL

This section aims to establish the regret bound of VR-MCL. Our method involves three key parts: Firstly, we define the objective form of Regret Bound in continual learning. Secondly, we prove that updating the objective of VR-MCL is equivalent to updating the latent optimal loss function, by bounding the gradient of VR-MCL with the latent optimal gradients. Finally, we derive the regret bound of VR-MCL based on the aforementioned steps.

**(Part I)** We first define the Regret Bound in continual learning.

**Definition 1** (The Regret Bound in CL). *Consider an online learning scenario where an agent makes a decision by choosing an action $\theta_t \in \mathbb{R}^d$ at each time step $t = 1, ..., T$, and experiences a corresponding loss $\mathcal{L}_t(\theta_t)$. The primary objective of the agent is to minimize the difference between its cumulative loss and that of the best action in hindsight, which is commonly known as the regret.*

$$\mathbf{R}_T = \sum_{t=1}^T \mathcal{L}_t(\theta_t) - \min_{\theta \in \Theta} \sum_{t=1}^T \mathcal{L}_t(\theta)$$

*Similarly, in MCL, for the current task $\mathcal{T}^j$, the goal is to minimize $\mathbf{CR}_j = \tilde{F}(\theta) - F(\theta^*)$, where*

$$\tilde{F}(\theta) := \sum_{i=1}^j \mathcal{L}^i(\hat{\theta}^j) \qquad and \qquad F(\theta^*) = \min_\theta \sum_{i=1}^j \mathcal{L}^i(\theta).$$

This proof is grounded on certain assumptions, which are listed below:

**Assumption 1.** *The compact convex set $\mathcal{C} \subseteq \mathbb{R}^d$ has diameter $D$, i.e., $\forall \theta, \theta' \in \mathcal{C}, \|\theta - \theta'\| \leq D$.*

**Assumption 2.** *The function $F(\cdot)$ is $G$-Lipschitz, $\mu$-strongly convex, $\varphi$-smooth (i.e., has $\varphi$-Lipschitz gradients) and has $\kappa$-Lipschitz hessian. Then, according to the Theorem 1 in Finn et al. (2019), when the step size $\alpha < \{\frac{1}{2\varphi}, \frac{\mu}{8\kappa G}\}$, the gradient $\nabla \tilde{F}(x, \epsilon_t)$ is $L$-Lipschitz ($L = \frac{9\varphi}{8}$) over the constraint set $\mathcal{C}$, that is*

$$\|\nabla \tilde{F}(\theta, \epsilon_t) - \nabla \tilde{F}(\theta', \epsilon_t)\| \leq L\|\theta - \theta'\| = \frac{9\varphi}{8}\|\theta - \theta'\|, \forall \theta, \theta' \in \mathcal{C}.$$

**Assumption 3.** *Let $\tilde{F}_t$ denote the $t$-th training step, then the distance between our computed $\nabla \tilde{F}_t(\theta, \epsilon_t)$ and the gradient derived from the optimal function is bounded over the constraint set $\mathcal{C}$, i.e., for any $\theta \in \mathcal{C}, t \in \{1, ..., T\}$, there exists $\sigma^2 < \infty$ such that with probability 1,*

$$\|\nabla \tilde{F}_t(\theta, \epsilon_t) - \nabla F(\theta)\|^2 \leq \sigma^2$$

**Assumption 4.** *Let $F_t(\theta)$ mean the optimal $t$-th training step, and the difference of $F_t(\theta)$ and $F(\theta)$ is bounded over the constraint set $\mathcal{C}$, i.e., $\forall \theta \in \mathcal{C}, t \in \{1, ..., T\}$, there exists $M^2 < \infty$ such that with probability 1,*

$$\|F_t(\theta) - F(\theta)\|^2 \leq M^2$$

**(Part II)** In the following, we will demonstrate that updating the objective function of VR-MCL can be regarded as updating the latent optimal loss function. This is achieved by bounding the gradient of VR-MCL with the latent optimal gradients. Specifically, the updating gradient can be expressed as $m_t = \nabla \tilde{F}(\theta_t, \epsilon_t) + r(m_{t-1} - \nabla \tilde{F}(\theta_{t-1}, \epsilon_t))$. Let $\alpha$ denote the small learning rate and define $\Delta_t = m_t - \nabla F(\theta_t)$, our aim is to prove that $\Delta_t$ is bounded. Here, we suppose the momentum ratio $r = (1 - \rho_t)$, the learning rate $\alpha = \eta_t$, and $\rho_t = \eta_t = 1/(t + 1)^\gamma, \gamma \in (0, 1]$.

**Lemma 2.** *If Assumptions 1,2,3 are satisfied, $\forall t \geq 1, \delta_0 \in (0, 1)$, we have with probability at least $1 - \delta_0$,*

$$\|\Delta_t\| \leq 2\left(2LD + \frac{3^\gamma \sigma}{3^\gamma - 1}\right)(t + 1)^{-\gamma/2}\sqrt{2\log(4/\delta_0)}$$

Lemma 2 demonstrates that the gradient approximation error $\|\Delta_t\|$ is bounded with high probability as $t$ increases.

*Proof.* We derive the formulation of $\Delta_t$ at first.

$$
\begin{aligned}
\Delta_t &= m_t - \nabla F(\theta_t) \\
&= (1-\rho_t)\Delta_{t-1} + (1-\rho_t)(\nabla \tilde{F}_t(\theta_t, \epsilon_t) - \nabla \tilde{F}_t(\theta_{t-1}, \epsilon_t) - (\nabla F(\theta_t) - \nabla F(\theta_{t-1}))) \\
&\quad + \rho_t(\nabla \tilde{F}_t(\theta_t, \epsilon_t) - \nabla F(\theta_t)) \\
&= \prod_{k=2}^{\tau}(1-\rho_k)\Delta_1 + \sum_{\tau=2}^{t}\prod_{k=\tau}^{t}(1-\rho_k)(\nabla \tilde{F}_\tau(\theta_\tau, \epsilon_\tau) - \nabla \tilde{F}_\tau(\theta_{\tau-1}, \epsilon_\tau) - (\nabla F(\theta_\tau) - \nabla F(\theta_{\tau-1}))) \\
&\quad + \sum_{\tau=2}^{t}\rho_\tau \prod_{k=\tau+1}^{t}(1-\rho_k)(\nabla \tilde{F}_t(\theta_\tau, \epsilon_\tau) - \nabla F(\theta_\tau))
\end{aligned}
$$

Let $\Delta_t = \sum_{\tau=1}^{t}\zeta_{t,\tau}$, where $\zeta_{t,1} = r^{t-1}\Delta_1$ and for $\tau > 1$, $\zeta_{t,\tau} = \prod_{k=\tau}^{t}(1-\rho_k)(\nabla \tilde{F}_\tau(\theta_\tau, \epsilon_\tau) - \nabla \tilde{F}_\tau(\theta_{\tau-1}, \epsilon_\tau) - (\nabla F(\theta_\tau) - \nabla F(\theta_{\tau-1}))) + \rho_\tau \prod_{k=\tau+1}^{t}(1-\rho_k)(\nabla \tilde{F}_\tau(\theta_\tau, \epsilon_\tau) - \nabla F(\theta_\tau))$. Recall $\Delta_1 = \nabla \tilde{F}(\theta_1, \epsilon_1) - \nabla F(\theta_1)$. We observe that $E\left[\zeta_{t,\tau}\|\mathcal{F}_{\tau-1}\right] = 0$ when $\mathcal{F}_{\tau-1}$ is the $\sigma$-field generate by $\{\mathcal{L}_1, \epsilon_1, ..., \mathcal{L}_{\tau-1}, \epsilon_{\tau-1}\}$. Therefore, $\{\zeta_{t,\tau}\}_{\tau=1}^{t}$ is a martingale difference sequence. Then, we will derive upper bounds of $\|\zeta_{t,\tau}\|$. We initially proved the following results,

$$
\prod_{k=\tau}^{t}(1-\rho_k) = \prod_{k=\tau}^{t}\left(1 - \frac{1}{(k+1)^\gamma}\right) = \prod_{k=\tau}^{t}\frac{(k+1)^\gamma - 1}{(k+1)^\gamma} \leq \prod_{k=\tau}^{t}\frac{k^\gamma}{(k+1)^\gamma} = \frac{\tau^\gamma}{(t+1)^\gamma},
$$

where the inequality holds from the concavity of $h(x) = x^\gamma$ for any $x \geq 0$. Based on this inequality we can bound $\|\zeta_{t,1}\|$ as follows,

$$
\|\zeta_{t,1}\| \leq \frac{2^\gamma}{(t+1)^\gamma}\left\|\nabla \tilde{F}_1(\theta_1, \epsilon_1), \nabla F(\theta_1)\right\| \leq \frac{2^\gamma \sigma}{(t+1)^\alpha} \stackrel{\text{def}}{=} c_{t,1},
$$

where the second inequality holds according to Assumption 3. When $\tau \geq 1$, we can get,

$$
\begin{aligned}
\|\zeta_{t,\tau}\| &\leq \prod_{k=\tau}^{t}(1-\rho_k)\left(\left\|\nabla \tilde{F}_\tau(\theta_\tau, \epsilon_\tau) - \nabla F_\tau(\theta_{\tau-1}, \epsilon_\tau)\right\| + \|\nabla F(\theta_\tau) - \nabla F(\theta_{\tau-1})\|\right) \\
&\quad + \rho_\tau \prod_{k=\tau+1}^{t}(1-\rho_k)\|\nabla \tilde{F}_\tau(\theta_\tau, \epsilon_\tau) - \nabla F(\theta_\tau)\| \\
&\leq 2L\|\theta_\tau - \theta_{\tau-1}\|\prod_{k=\tau}^{t}(1-\rho_k) + \sigma\rho_\tau \prod_{k=\tau+1}^{t}(1-\rho_k) \\
&\leq 2LD\rho_{\tau-1}\prod_{k=\tau}^{t}(1-\rho_k) + \sigma\rho_\tau \prod_{k=\tau+1}^{t}(1-\rho_k)
\end{aligned} \tag{4}
$$

We can further derive the final term as follows,

$$
\begin{aligned}
\rho_\tau \prod_{k=\tau+1}^{t}(1-\rho_k) &= \frac{\rho_\tau}{\rho_{\tau-1}(1-\rho_\tau)}\left(\rho_{\tau-1}\prod_{k=\tau}^{t}(1-\rho_k)\right) \leq \frac{1}{1-\rho_\tau}\left(\rho_{\tau-1}\prod_{k=\tau}^{t}(1-\rho_k)\right) \\
&\leq \frac{1}{1-1/3^\gamma}\left(\rho_{\tau-1}\prod_{k=\tau}^{t}(1-\rho_k)\right) \leq \frac{3^\gamma}{3^\gamma - 1}\left(\rho_{\tau-1}\prod_{k=\tau}^{t}(1-\rho_k)\right).
\end{aligned} \tag{5}
$$

By plugging Eqn. (5) into Eqn. (4, we can get, $\forall \tau \geq 1$

$$
\|\zeta_{t,\tau}\| \leq (2LD + \frac{3^\gamma \sigma}{3^\gamma - 1})\rho_{\tau-1}\prod_{k=\tau}^{t}(1-\rho_k) \stackrel{\text{def}}{=} c_{t,\tau}
$$

According to Theorem 3.5 in Pinelis (1994), we can get $\forall \lambda \geq 0$,

$$
\mathbb{P}\left(\|\Delta_t\| \geq \lambda\right) \leq 4\exp\left(-\frac{\lambda^2}{4\sum_{\tau=1}^{t}c_{t,\tau}^2}\right), \tag{6}
$$

where $\mathbf{c_{t,1}}$ is defined in Eqn. (A.4) and $\mathbf{c_{t,\tau}}, (\tau \geq 1)$ is defined in Eqn. (4). Then we can get,

$$\sum_{\tau=1}^{t} c_{t,\tau}^2 = c_{t,1}^2 + \sum_{\tau=2}^{t} c_{t,\tau}^2 = \frac{2^{2\gamma}\sigma^2}{(t+1)^{2\gamma}} + \left(2LD + \frac{3^\gamma \sigma}{3^\gamma - 1}\right)^2 \sum_{\tau=2}^{t} \left(\rho_{\tau-1} \prod_{k=\tau}^{t} (1-\rho_k)\right)^2$$

$$\leq \frac{2^{2\gamma}\sigma^2}{(t+1)^{2\gamma}} + \frac{\left(2LD + \frac{3^\gamma \sigma}{3^\alpha - 1}\right)^2}{(t+1)^\gamma} \leq \frac{((\sqrt{2})^\gamma \sigma)^2}{(t+1)^\gamma} + \frac{\left(2LD + \frac{3^\gamma \sigma}{3^\gamma - 1}\right)^2}{(t+1)^\gamma} \leq \frac{2\left(2LD + \frac{3^\gamma \sigma}{3^\gamma - 1}\right)^2}{(t+1)^\gamma},$$

(7)

where the last inequality is because $(\sqrt{2})^\gamma \leq 3^\gamma/(3^\gamma - 1) \; \forall \gamma \in (0,1]$. Substituting Eqn. (7) into Eqn. (6) and setting $\lambda = 2\left(2LD + \frac{3^\gamma \sigma}{3^\gamma-1}\right)(t+1)^{-\gamma/2}\sqrt{2\log(4/\delta_0)}$ for some $\delta_0 \in (0,1)$, we have with probability at least $1 - \delta_0$,

$$\|\Delta_t\| \leq 2\left(2LD + \frac{3^\gamma \sigma}{3^\gamma-1}\right)(t+1)^{-\gamma/2}\sqrt{2\log(4/\delta_0)},$$

which is the expected result. $\qquad\square$

(**Part III**) Next, we will establish the regret bound $\mathbf{CR_j}$ for continual learning.

**Theorem 1** (Regret Bound of VR-MCL). *If $F$ is convex and all aforementioned four Assumptions are satisfied, then with probability at least $1 - \delta$ for any $\delta \in (0,1)$,*

$$\mathbf{CR}_j \leq (\log T + 1)(F(\theta_1) - F(\theta^*)) + \frac{LD^2(\log T + 1)^2}{2}$$
$$+ \frac{LD^2(\log T + 1)^2}{2} + \left(16LD^2 + 16\sigma D + 4M\right)\sqrt{2T\log(8T/\delta)} = \tilde{\mathcal{O}}(\sqrt{T})$$

*Proof.* We initially define a sequence $s_t = \tilde{F}_t(\theta_t) - F_t(\theta^*) - (F(\theta_t) - F(\theta^*)), t = 1, ..., T$. It can be observed that $\mathbb{E}\left[s_t \mid \mathcal{F}_{t-1}\right] = 0$, where $\mathcal{F}_{t-1}$ is the $\sigma$-algebra generated by $\{F_1, \epsilon_1, ..., F_{t-1}, \epsilon_{t-1}\}$. It means that $\{s_t\}_{t=1}^{T}$ is a martingale difference sequence. According to Assumption 4, we have

$$\|s_t\| = \|\tilde{F}_t(\theta_t) - F_t(\theta^*) - (F(\theta_t) - F(\theta^*))\| \leq 2M$$

According to Theorem 3.5 in Pinelis (1994), we can get

$$\mathbb{P}\left(\|\sum_{t=1}^{T} s_t\| \geq \lambda\right) \leq 4\exp\left(-\frac{\lambda^2}{16TM^2}\right),$$

where $\lambda > 0$. By setting $\lambda = 4M\sqrt{T log(8/\delta)}$, we have with probability at least $1 - \delta/2$,

$$\sum_{t=1}^{T} s_t = \sum_{t=1}^{T}(\tilde{F}_t(\theta_t) - F_t(\theta^*)) - \sum_{t=1}^{T}(F(\theta_t) - F(\theta^*)) \leq 4M\sqrt{T log(8/\delta)}.$$

By rearranging the above terms, we get

$$\mathbf{CR}_j = \tilde{F}(\theta) - F(\theta^*) = \sum_{t=1}^{T}(\tilde{F}_t(\theta_t) - F_t(\theta^*)) \leq \sum_{t=1}^{T}(F(\theta_t) - F(\theta^*)) + 4M\sqrt{T log(8/\delta)}. \quad (8)$$

Then according to Lemma 2 in (Mokhtari et al., 2020), we can derive the first term in Eqn. (8) as follows,

$$
\begin{aligned}
F(\theta_t) - F(\theta^*) &= (1 - \eta_t)(F(\theta_{t-1}) - F(\theta^*)) + \eta_t D\|\Delta_{t-1}\|^2 + \frac{LD^2\eta_t^2}{2} \\
&\leq \prod_{\tau=1}^{t-1}(1 - \eta_\tau)(F(\theta_1) - F(\theta^*)) + \sum_{\tau=1}^{t-1}\eta_\tau(D\|\Delta_\tau\|^2 + \frac{LD^2\eta_\tau}{2}) \prod_{k=\tau+1}^{t-1}(1 - \eta_k) \\
&= \frac{1}{t}(F(\theta_1) - F(\theta^*)) + \sum_{\tau=1}^{t-1}\frac{1}{\tau+1}(D\|\Delta_\tau\|^2 + \frac{LD^2}{2(\tau+1)})\frac{\tau+1}{t} \\
&= \frac{1}{t}(F(\theta_1) - F(\theta^*)) + \frac{1}{t}\sum_{\tau=1}^{t-1}(D\|\Delta_\tau\|^2 + \frac{LD^2}{2(\tau+1)}) \\
&\leq \frac{1}{t}(F(\theta_1) - F(\theta^*)) + \frac{D}{t}\sum_{\tau=1}^{t-1}\|\Delta_\tau\|^2 + \frac{LD^2\log t}{2t}
\end{aligned}
\tag{9}
$$

Summing Eqn. (9) from $t = 1$ to $T$, we obtain,

$$
\begin{aligned}
\sum_{t=1}^{T}F(\theta_t) - F(\theta^*) &\leq \sum_{t=1}^{T}\frac{1}{t}(F(\theta_1) - F(\theta^*)) + \sum_{t=1}^{T}\sum_{\tau=1}^{t-1}\frac{D}{t}\|\Delta_\tau\|^2 + \sum_{t=1}^{T}\frac{LD^2\log t}{2t} \\
&\leq \sum_{t=1}^{T}\frac{1}{t}(F(\theta_1) - F(\theta^*)) + \sum_{t=1}^{T}\sum_{\tau=1}^{t-1}\frac{D}{t}\|\Delta_\tau\|^2 + \frac{LD^2\log T}{2}\sum_{t=1}^{T}\frac{1}{t} \\
&\leq (\log T + 1)(F(\theta_1) - F(\theta^*)) + \sum_{t=1}^{T}\sum_{\tau=1}^{t-1}\frac{D}{t}\|\Delta_\tau\|^2 + \frac{LD^2(\log T + 1)^2}{2}
\end{aligned}
\tag{10}
$$

According to Lemma 2, we can obtain the second term is,

$$
\begin{aligned}
\sum_{t=1}^{T}\sum_{\tau=1}^{t-1}\frac{D}{t}\|\Delta_\tau\| &\leq 4(LD^2 + \sigma D)\sqrt{2\log(8T/\sigma)}\sum_{t=1}^{T}\sum_{\tau=1}^{t-1}\frac{1}{t\sqrt{\tau+1}} \\
&\leq 4(LD^2 + \sigma D)\sqrt{2\log(8T/\sigma)}\sum_{t=1}^{T}\frac{2\sqrt{t}}{t} \\
&\leq 16(LD^2 + \sigma D)\sqrt{2T\log(8T/\sigma)}
\end{aligned}
\tag{11}
$$

Substituting Eqn. (11) into Eqn. (10), we have with probability at least $1 - \delta/2$,

$$
\sum_{t=1}^{T}F(\theta_t) - F(\theta^*) \leq \log(T+1)(F(\theta_1) - F^*(\theta)) + \frac{LD^2(\log T + 1)^2}{2} + 16(LD^2 + \sigma D)\sqrt{2T\log(8T/\delta)}
\tag{12}
$$

Combining Eqn. (12) with Eqn. (8) we have with probability at least $1 - \delta$,

$$
\begin{aligned}
\mathbf{CR}_j &\leq (\log T + 1)(F(\theta_1) - F(\theta^*)) + \frac{LD^2(\log T + 1)^2}{2} \\
&+ \frac{LD^2(\log T + 1)^2}{2} + (16LD^2 + 16\sigma D + 4M)\sqrt{2T\log(8T/\delta)} = \tilde{\mathcal{O}}(\sqrt{T})
\end{aligned}
$$

$\square$

## B  RELATED WORKS

**Regularization-based methods (or parameter regularization methods)** usually construct a quadratic regularization term to slow down the learning of weights that are important to prior tasks. As we know, the Fisher information matrix is equivalent to the Hessian matrix if the loss function

is negative log-likelihood and we get a ground truth probabilistic model (Chapter 4.5 of (Keener, 2010)). In this way, EWC (Kirkpatrick et al., 2017) and on-EWC (Huszár, 2017), as the first work applying parameter regularization, utilize the diagonal Fisher information matrix to approximate the Hessian matrix, while Kronecker factored Laplace approximation (KFLA) (Ritter et al., 2018) exploits Kronecker factored Laplace to compute the Fisher information matrix with off-diagonal elements. To account for the trajectory of model training, IS (Zenke et al., 2017) proposes an online estimation of a diagonal matrix and assigns weights based on its contribution to loss decay. On the other hand, in class-incremental learning, IADM (Yang et al., 2019) and CE-IDM (Yang et al., 2021) report that different layers have varying characteristics. Shallow layers with limited representation tend to converge faster, while deep layers with powerful discrimination abilities tend to converge more slowly. Therefore, based on EWC (Kirkpatrick et al., 2017), IADM (Yang et al., 2019) learns an online layer-wise importance matrix.

**Rehearsal-based methods** address catastrophic forgetting by replaying previous task samples from a memory buffer. Experience Replay (ER) (Rolnick et al., 2019) directly sample data from previous tasks and put them jointly train with current data. On the basis of ER, recent studies have further extended this idea. Meta Experience Replay (MER) (Riemer et al., 2019) views replay as a meta-learning problem for maximizing the transfer from previous tasks and minimizing the interference. Gradient-based Sample Selection (GSS) (Aljundi et al., 2019) pays attention to the samples stored in the memory module and hopes to increase the diversity of samples in the gradient space. Gradient Episodic Memory (GEM) (Lopez-Paz & Ranzato, 2017) and its lightweight variant Averaged-GEM (A-GEM) (Chaudhry et al., 2018) formulate optimization constraints such that gradients between current and old training data are aligned to further determine the final direction of optimization. Dark Experience Replay (DER++) (Buzzega et al., 2020) adds the knowledge distillation part on top of ER to regularize the logits of samples stored in the memory buffer. CLSER (Arani et al., 2021) introduces an innovative approach to episodic replay (ER) by presenting a dual memory framework. This framework incorporates both short-term and long-term semantic memories, allowing for their interaction with the episodic memory. CBA (Wang et al., 2023a) proposes to address the issue of recency bias by introducing a specially designed bias attractor, guided by the memory buffer $\mathcal{M}$.

**Meta-Continual Learning** (Meta-CL) is a class of methods that apply meta-learning (Finn et al., 2017; Wu et al., 2022; 2021; Wang et al., 2023b) in continual learning. From the bi-level optimization perspective to analyze meta-learning, its inner-loop (or lower-level) optimization usually fits the training data, while the outer-loop (higher-level) centralized test data aims to increase the model's generalization ability. Therefore, meta-learning is well suited for continual learning, as we want the model to fit the current task and still perform well on all observed tasks (i.e., previous and current tasks). Meta Experience Replay (MER) (Riemer et al., 2019), inspired by GEM (Lopez-Paz & Ranzato, 2017), utilizes replay to incentivize the alignment of gradients between old and new tasks to further maximize the transfer from previous tasks and minimize the interference. Online-aware Meta Learning (OML) (Javed & White, 2019) adopts a pre-training algorithm to learn an optimal representation offline, which is frozen when training in the downstream tasks. However, this offline training manner and assumption that the training data is a correlated data stream limit its applicability to practical scenarios. Besides, MER (Riemer et al., 2019) and OML (Javed & White, 2019) use a single sample at one time during the inner-loop optimization, which is time-consuming and not suitable for large-scale CL. To alleviate these problems, Look-ahead Meta learning (La-MAML) (Gupta et al., 2020) proposed an online gradient-based meta-learning algorithm and proved it is equivalent to a simplified version of gradient alignment. On the other hand, some Meta-CL algorithms (Rajasegaran et al., 2020; 2022) focus on combining meta-learning with task-agnostic continual learning that does not require task information during training and testing. Besides, the variance reduction has also been integrated with first-order meta-learning (Yang & Kwok, 2022; Wang et al., 2021). In this work, we focus on the *online* continual learning setting and provide a variance reduction Meta-CL (second-order) based on our novel understanding of Meta-CL, it also links Meta-CL with regularization-based methods.

**Online and Imbalanced Continual learning**, different from the traditional offline CL setting, are more challenging CL settings that have not been explored as much. For **online CL** scenarios, the initial methods still have been developed based on similar ways in offline CL (Lopez-Paz & Ranzato, 2017; Chaudhry et al., 2018; Aljundi et al., 2019). Recently, there has been increasing interest in developing online CL methods that can fully utilize the single-pass data stream to decrease the performance difference between online and offline CL. OCDVC (Gu et al., 2022) advocates for

the dual view consistency strategy, while SDP (Koh et al.) utilizes knowledge distillation to fully leverage the data stream. OBC (Chrysakis & Moens, 2023) proposes a simple approach to mitigate the bias of online CL by changing the optimization manner of the output layer model. OCM (Guo et al., 2022) combines these motivations and aims to reduce feature bias and fully utilize streaming data through mutual information maximization. However, none of these methods directly focus on the performance drop between online and offline CL and analyze the underlying reasons, which is a motivation of this work.

For **imbalanced CL**, where the sample number of each task is different (Lai et al., 2022b;a), most works aim to directly design a sampling strategy that can balance the distribution of samples retrieved from the memory buffer that stores data from previous tasks. CBRS (Chrysakis & Moens, 2020) introduced a novel mechanism for class-balancing reservoir sampling based on the sample distribution within the buffer. PRS (Kim et al., 2020) utilizes current data stream statistics to determine the sample-in and sample-out process, maintaining a balanced replay memory. Additionally, InfoRS (Sun et al.) introduces a stochastic information theoretic reservoir sampler to select and store the informative samples. Coresets (Borsos et al., 2020) employs the bi-level optimization technique for the same purpose. Different from these methods, our proposed approach takes a different perspective. It aims to conduct a comprehensive analysis of the factors that contribute to the substantial performance drop observed in both imbalanced and online settings compared to offline learning. By understanding these underlying factors, we propose the VR-MCL to the model performance in imbalanced CL scenarios.

**Variance Reduction** is a widely used technique for reducing the variance in stochastic gradients and accelerating the optimization process. Early research on variance reduction methods, such as SAG Gower et al. (2020), SAGA (Defazio et al., 2014), and SVRG (Johnson & Zhang, 2013), was primarily focused on strongly convex optimization problems. However, in recent years, these methods have been extended to general non-convex optimization problems (Allen-Zhu & Hazan, 2016; Nguyen et al., 2017; Fang et al., 2018). Nonetheless, these methods typically require the expensive computation of the full-batch gradient by processing all samples in a task, which is impractical in the context of online continual learning where samples are processed in mini-batches and the complete sample set is unavailable. To alleviate this expensive computation problem, momentum-based variance reduction algorithms (Cutkosky & Orabona, 2019; Khanduri et al., 2021) are proposed. Specifically, these methods incorporate a momentum term to compute the stochastic gradients at two consecutive iterations on the same set of stochastic samples, thereby replacing the time-consuming computation of the full-batch gradient. Additionally, these methods have the same asymptotic convergence rate as other variance reduction methods.

## C  IMPLEMENTATION DETAILS

### C.1  EXPERIMENTAL SETTINGS

**Imbalanced Settings.** We conducted imbalanced online continual learning experiments on Seq-cifar10, consisting of five tasks, each with two classes. As shown in Fig. 4, when the imbalance ratio $\gamma = 2$, the number of samples for each class across all tasks is [5000, 4629, 4286, 3968, 3674, 3401, 3149, 2916, 2700, 2500]. We also considered the reversed version, where the number of samples for each class is [2500, 2700, 2916, 3149, 3401, 3674, 3968, 4286, 4629, 5000]. Without loss of generality, we additionally sampled a random version with [2700, 2500, 5000, 4286, 3674, 3968, 3149, 3401, 2916, 4629] samples per class across all tasks.

**Baselines.** We compared the proposed VR-MCL to various baseline approaches, including regularization-based methods, rehearsal-based methods, meta-continual learning methods, and specific CL methods designed for imbalanced and online settings.

**Regularization-based methods:**

- **On-EWC:** (Huszár, 2017). It approximates each Hessian matrix by a diagonal Fisher matrix calculated at the end of each task training. The larger Fisher information value means the corresponding weights are more important to the old tasks and thus should undergo less change in future tasks.

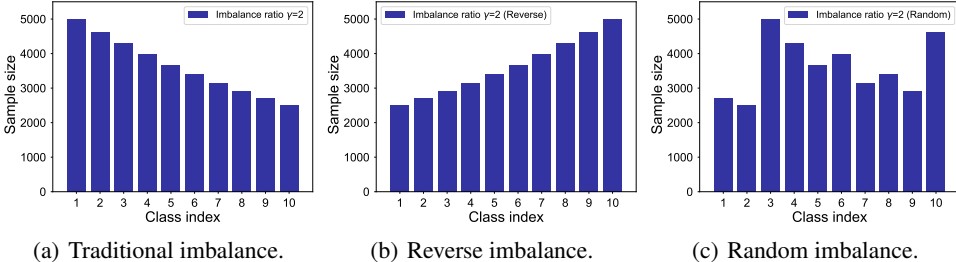

(a) Traditional imbalance.   (b) Reverse imbalance.   (c) Random imbalance.

Figure 4: Different imbalance settings of Seq-CIFAR10, where the imbalance ratio $\gamma = 2$.

- **IS:** (Zenke et al., 2017). It introduces the intelligent synapse. Each synapse accumulates task-relevant information over time so as to rapidly store new memories without forgetting the learned ones. From the Hessian approximation perspective, it constructs a diagonal regularization matrix (i.e., a generalization form of the Hessian matrix integrating over time) to estimate the Hessian matrix.
- **LWF:** (Li & Hoiem, 2017). It is similar to joint training, but the key difference is that LWF does not require data and labels from the old tasks, instead employing distillation techniques.

**Rehearsal methods:**

- **GEM and A-GEM** (Lopez-Paz & Ranzato, 2017; Chaudhry et al., 2018). They formulate optimization constraints such that gradients between current and old training data are aligned to determine the final direction of optimization.
- **ER** (Rolnick et al., 2019). Training current task with data sampled from the memory buffer which stores the examples from previous tasks.
- **DER/DER++** (Buzzega et al., 2020). Adding the distillation loss of logits on the top of ER.
- **CLSER** (Arani et al., 2021). The dual memory experience replay (ER) method maintains short-term and long-term semantic memories that interact with episodic memory.

**CL methods focus on imbalance and online settings:**

- **CBRS** (Chrysakis & Moens, 2020). It introduces a novel mechanism for class-balancing reservoir sampling based on the sample distribution within the buffer to ensure more balanced samples are stored.
- **Coresets** (Borsos et al., 2020). It presents a novel coreset construction technique through cardinality-constrained bilevel optimization, which is effective in the imbalanced CL setting.
- **OCM** (Guo et al., 2022). It proposes to reduce feature bias and fully exploit streaming data through mutual information maximization. **Note that**, for a fair comparison, we choose its OCM (no local augmentation) for evaluation.
- **ER-OBC** (Chrysakis & Moens, 2023). It proposes a simple approach to mitigate bias in online CL by modifying the optimization process of the output layer model.

**Meta-CL methods:**

- **MER** (Riemer et al., 2019). It utilizes replay examples to incentivize the alignment of gradients between old and new tasks to further maximize the transfer from previous tasks and minimize interference.
- **La-MAML** (Gupta et al., 2020). It proposed an online gradient-based meta-learning algorithm and proved it is equivalent to a simplified version of MER. **Note that** in the online scenario of La-MAML (Gupta et al., 2020), the mini-batch samples instead of the whole task could be trained multiple times, which is not the strict online continual learning. In our experiments, all methods follow the standard online setting where all samples can only be trained once.

**Hyper-parameters.** The hyperparameters used for each experimental setting are listed in Table 7. Let $lr$ denote the learning rate, $bs$ denote the batch size and $mbs$ means the minibatch size (i.e., the batch size drawn from the buffer). It is worth noting that we used the same hyperparameters across different datasets and buffer sizes, which shows our method has good generalizability without requiring extensive hyperparameter tuning for each specific setting.

Table 7: Hyperparameters on all the three datasets including Seq-Cifar10/100 and Seq-TinyImgaeNet.

| Backbone | Epochs | Buffer size | Inner lr ($\beta$) | Outer lr ($\alpha$) | r | inner-bs | bs | mbs |
|---|---|---|---|---|---|---|---|---|
| PcCNN | 1 | 200 | 0.1 | 0.25 | 0.25 | 2 | 10 | 10 |
| PcCNN | 1 | 600 | 0.1 | 0.25 | 0.25 | 2 | 10 | 10 |
| PcCNN | 1 | 1000 | 0.1 | 0.25 | 0.25 | 2 | 10 | 10 |
| Reduced ResNet-18 | 1 | 200 | 0.1 | 0.25 | 0.25 | 8 | 32 | 32 |
| Reduced ResNet-18 | 1 | 600 | 0.1 | 0.25 | 0.25 | 8 | 32 | 32 |
| Reduced ResNet-18 | 1 | 1000 | 0.1 | 0.25 | 0.25 | 8 | 32 | 32 |

## C.2 THE MASK TRAINING OF THE PROPOSED VR-MCL

For the implementation of the VR-MCL, we adopt mask training different from La-MAML (i.e., one mask training strategy on the final output layer). In this subsection, we will first introduce two mask training approaches: Mask-LM, which is the original mask training in La-MAML, and Mask-VR, which is the proposed mask training used in VR-MCL. We will then provide both mathematical and experimental analysis to compare the two mask training methods.

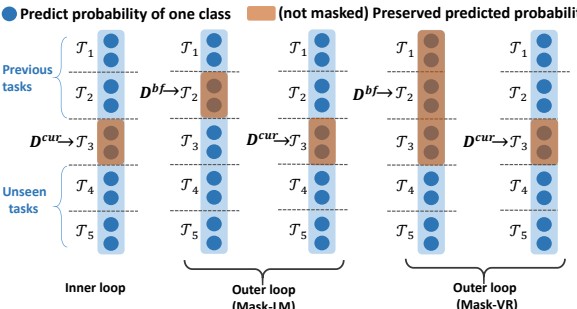

Figure 5: The illustration of the two mask training approaches, where the current training task is $\mathcal{T}^3$.

**Introduction of the two Mask training methods** As the output layer is shared among all tasks, meta-continual learning methods have the problem of a mismatch between the limited rehearsal data from previous tasks $\mathcal{T}^{[1:j-1]}$ and the larger incoming data from the current task $\mathcal{T}^j$. To solve this problem, a masked softmax loss is applied by La-MAML Gupta et al. (2020). As shown in Figure 5, La-MAML applies a mask (i.e., Mask-LM) that only keeps the logits of the corresponding classes within a task to both the loss in the inner-loop adaptation and outer-loop optimization. The Mask-LM improves the inner-loop adaptation for the current tasks and mitigates the interference between the limited data sampled from the memory buffer $\mathcal{M}$ and the larger incoming data from the current task. However, the employment of the Mask-LM in the outer-loop still has a limitation. The left part of Mask-LM concentrates solely on minimizing the intra-task loss, without considering the interdependencies between different classes from various tasks. This oversight significantly reduces the model's efficacy in the class incremental setting. To address this issue, we propose a modification (i.e., Mask-VR) inspired by (Caccia et al., 2021), where we mask all seen classes that appeared in the memory buffer $\mathcal{M}$. This modification enables the model to distinguish samples belonging to different tasks, which is shown in the left part of Mask-VR. **Mathematical and Experimental Analysis.** The main idea of Mask-VR to enlarge the preserved logits is to introduce the negative gradients so as to get more concise class prototypes and thus achieve better performance. To investigate the impact of the proposed Mask-VR, we conducted experiments and demonstrated empirically that it can effectively reduce the gradient variance. Furthermore, our findings suggest that the Mask-VR and the proposed variance reduction method can be synergistically combined to effectively reduce the gradient variance.

We will begin by conducting the **mathematical analysis**. Note that the following mathematical symbols are solely for the purpose of illustrating the Mask-VR, they are not aligned with the mathematical symbols used in the main text.

Let $C_{seen}$ refer to all seen classes, and $|C_{seen}|$ is the number of classes the model has seen. In Figure 5, the current training task is $\mathcal{T}^3$ and $|C_{seen}| = 6$. Let the model architecture consists of two parts: feature extractor $h_\theta$ and classifier $f_\phi$ and the output of the model before the softmax operation is $o(x; \theta, \phi) = f_\phi(h_\theta(x))$. Then the cross-entropy loss in continual learning $\mathcal{L}_{ce}$ is,

$$\mathcal{L}_{ce}(o(x; \theta, \phi)) = -\sum_{i=1}^{|C_{seen}|} l_{c_i} log(p^{c_i}), \qquad p^{c_i} = \frac{exp(o_{c_i})}{\sum_{s=1}^{|C_{seen}|} exp(o_{c_s})}$$

where $x$ is the training sample, $l_{c_i} \in \{0, 1\}$ is the one-hot label of class $c_i$, and $o(x; \theta, \phi) = [o_{c_1}, o_{c_2}, ..., o_{c_{|C_{seen}|}}]$ is the corresponding logit values of $x$. Let $x^i$ mean the sample of class $c_i$, then the gradients on the classifier $f_\phi$ is,

$$\frac{\partial \mathcal{L}_{ce}\left(o\left(x^i; \theta, \phi\right)\right)}{\partial o_{c_i}} = p^{c_i} - 1, \qquad \frac{\partial \mathcal{L}_{ce}\left(o\left(x^i; \theta, \phi\right)\right)}{\partial o_{c_j}} = p^{c_j},$$

$$\frac{\partial \mathcal{L}_{ce}\left(o\left(x^i; \theta, \phi\right)\right)}{\partial \phi_{c_i}} = \frac{\partial \mathcal{L}_{ce}\left(o\left(x^i; \theta, \phi\right)\right)}{\partial o_{c_i}} \frac{\partial o_{c_i}}{\partial \phi_{c_i}} = (p^{c_i} - 1) h_\theta(x^i), \quad \text{[Positive gradient]}$$

$$\frac{\partial \mathcal{L}_{ce}\left(o\left(x^i; \theta, \phi\right)\right)}{\partial \phi_{c_j}} = \frac{\partial \mathcal{L}_{ce}\left(o\left(x^i; \theta, \phi\right)\right)}{\partial o_{c_j}} \frac{\partial o_{c_j}}{\partial \phi_{c_j}} = p^{c_j} h_\theta(x^i), \quad \text{[Negative gradient]}$$

Let $\mathbf{W}$ denote the matrix with all class prototypes $\{\mathbf{w}_c\}_{c \in C_{seen}}$, then $\mathbf{W} \circ h_\theta = f_\phi(h_\theta(x))$. According to the above equation, we can get in Mask-LM, only $\mathbf{w}_{c_i}$ receives a positive gradient from $x^i$, and there are no negative gradients due to preserving only a portion of the logits. However, in Mask-VR, both positive and negative gradients are present, similar to the concept of positives and negatives in contrastive loss. Consequently, the class prototypes $\mathbf{W}$ learned with Mask-VR exhibit superior performance compared to those learned with Mask-LM.

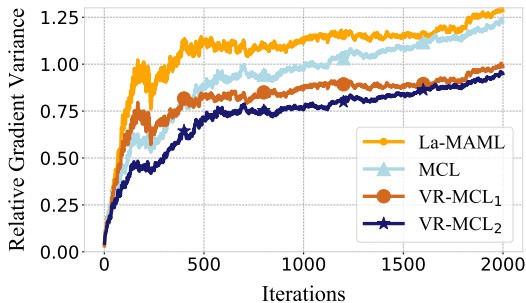

Figure 6: Relative gradient variance of update during training. The subscript 1 represents training with Mask-LM, while the subscript 2 represents training with Mask-VR.

**Ablation studies of the Mask-Training.** To better understand the effectiveness of the Mask-VR, and the proposed VR-MCL, we compare the performance of La-MAML, MCL(La-MAML with Mask-VR), as well as VR-MCL$_1$ (i.e., VR-MCL with Mask-LM) and VR-MCL$_2$ (i.e., VR-MCL with Mask-VR), as shown in Table 8. Moreover, we also plot the relative gradient variance of La-MAML, MCL, VR-MCL$_1$, and VR-MCL$_2$, as shown in Fig. 6. Based on the fact that the curves with variance reduction (VR) are all lower than those without VR, it can be concluded that: 1) the proposed VR technique indeed can reduce the variance, 2) the effects of VR and Mask-VR can be superimposed and work together.

Table 8: Performance of La-MAML, MCL, VR-MCL$_1$ and VR-MCL$_2$ on Seq-CIFAR10.

| Method | La-MAML | MCL | VR-MCL$_1$ | VR-MCL$_2$ |
|---|---|---|---|---|
| Acc | $33.43_{\pm 1.21}$ | $52.40_{\pm 2.13}$ | $35.66_{\pm 1.13}$ | $\mathbf{56.48}_{\pm \mathbf{1.79}}$ |
| AAA | $42.98_{\pm 1.60}$ | $65.87_{\pm 1.26}$ | $50.02_{\pm 1.80}$ | $\mathbf{66.97}_{\pm \mathbf{1.58}}$ |

# D    OTHER EXPERIMENTAL RESULTS.

## D.1    THE TRAINING TIME ANALYSIS OF VR-MCL

To analyze the time complexity of the proposed algorithm, we compared the training time of the proposed VR-MCL with relevant Meta-CL methods on the Seq-CIFAR-10 dataset using reduced ResNet-18, as shown in Table 9. It is evident that our proposed VR-MCL method demonstrates a substantial reduction in training time compared to MER. To assess the value of the additional training time required by VR-MCL, we conducted experiments by extending the training epoch of La-MAML from 1 to 2. Surprisingly, despite the comparable training time with VR-MCL, La-MAML achieved a performance of only 38.89%, which is extremely lower than the performance of VR-MCL (i.e., 56.48%). This outcome serves as strong evidence for the superiority of VR-MCL in terms of both training efficiency and performance on Seq-CIFAR10.

Table 9: Training time comparison of different meta-continual learning methods on Seq-CIFAR10 under the online setting (i.e., the training epoch of each task is set as one.)

| Method | La-MAML | MER | VR-MCL |
|---|---|---|---|
| Training Epochs | 1 | 1 | 1 |
| Training Time (s) | 750.61 | 20697.70 | 1297.10 |
| Training Epochs | 2 | 1 | 1 |
| Training Time (s) | 1511.54 | 20697.70 | 1297.10 |
| AAA | 53.21% | 50.99% | **66.97%** |
| Acc | 38.89% | 36.92% | **56.48%** |

## D.2    OTHER EVALUATION METRICS

To further evaluate the effectiveness of the proposed VR-MCL, we provide the backward transfer (BWT) metric in Table 10, which measures the performance degradation in subsequent tasks. It can be observed our VR-MCL achieves the best results. It is worth noting that in Table 10, La-MAML is excluded from the analysis due to the extremely negative impact of Mask-LM utilization on its newest training task (i.e., $\mathcal{T}^j$) performance (Caccia et al., 2021). This impact results in large and even a positive backward transfer (BWT) value for La-MAML.

Table 10: The Backward Transfer of Seq-CIFAR10 and longer task sequences Seq-CIFAR100, Seq-TinyImageNet with 95% confidence interval on reduced ResNet-18 under the **online** setting. The memory buffer size is set as 1000 (i.e., $|\mathcal{M}|$=1000). All reported numbers are the average of 5 runs. Shaded areas are our methods, and '−' indicates the result was omitted due to high instability.

| Method | CIFAR-10 BWT | CIFAR-100 BWT | Tiny-ImageNet BWT |
|---|---|---|---|
| SGD | $-61.48_{\pm 1.58}$ | $-44.53_{\pm 0.38}$ | $-34.59_{\pm 0.35}$ |
| On-EWC | $-65.94_{\pm 2.08}$ | $-42.45_{\pm 1.30}$ | $-33.05_{\pm 0.56}$ |
| IS | $-63.38_{\pm 1.19}$ | $-39.54_{\pm 1.58}$ | $-23.76_{\pm 1.27}$ |
| A-GEM | $-67.06_{\pm 1.11}$ | $-48.74_{\pm 0.37}$ | $-37.93_{\pm 1.04}$ |
| GEM | $-62.72_{\pm 2.00}$ | $-42.27_{\pm 1.90}$ | $-36.21_{\pm 0.79}$ |
| ER | $-33.97_{\pm 7.07}$ | $-31.24_{\pm 1.86}$ | $-31.07_{\pm 1.91}$ |
| DER | $-62.17_{\pm 3.48}$ | $-50.21_{\pm 1.32}$ | $-39.50_{\pm 0.60}$ |
| DER++ | $-24.65_{\pm 2.21}$ | $-46.74_{\pm 1.13}$ | $-39.16_{\pm 1.34}$ |
| CLSER | $-23.61_{\pm 1.54}$ | $-37.02_{\pm 1.35}$ | $-34.58_{\pm 0.85}$ |
| OCM | $-22.02_{\pm 1.92}$ | $-17.91_{\pm 0.74}$ | $-11.99_{\pm 0.86}$ |
| ER-OBC | $-34.71_{\pm 0.81}$ | $-5.39_{\pm 1.83}$ | $-6.20_{\pm 1.27}$ |
| MER | $-28.77_{\pm 1.25}$ | − | − |
| VR-MCL | $-20.92_{\pm 1.94}$ | $\mathbf{-0.16_{\pm 0.10}}$ | $\mathbf{-2.86_{\pm 0.83}}$ |

## D.3 HYPERPARAMETER SELECTION

We utilized a grid search technique to select the optimal hyperparameters, including the learning rate $\alpha$ and the momentum ratio $r$, as illustrated in Table 11 and Table 12. The results demonstrate that our proposed method achieves the best performance when both the momentum ratio $r$ and learning rate $\alpha$ are set to 0.25. Moreover, it is worth noting that there is small performance difference observed when different hyperparameters are chosen. This suggests that the proposed VR-MCL method exhibits robustness and is not highly sensitive to the selection of hyperparameters.

Table 11: Reduced ResNet18 performance with different momentum ratios $r$ on Seq-CIFAR10, using a fixed learning rate ($\alpha = 0.25$).

| $r$ | 0.15 | 0.25 | 0.35 |
|---|---|---|---|
| Acc | $55.46_{\pm 0.44}$ | $\mathbf{56.48}_{\pm \mathbf{1.79}}$ | $54.14_{\pm 1.50}$ |
| AAA | $66.46_{\pm 1.23}$ | $\mathbf{66.97}_{\pm \mathbf{1.58}}$ | $63.92_{\pm 0.89}$ |

Table 12: Reduced ResNet18 performance with different learning rates $\alpha$ on Seq-CIFAR10, using a fixed momentum ratio ($r = 0.25$).

| $\alpha$ | 0.15 | 0.25 | 0.35 |
|---|---|---|---|
| Acc | $54.466_{\pm 1.94}$ | $\mathbf{56.48}_{\pm \mathbf{1.79}}$ | $55.91_{\pm 1.52}$ |
| AAA | $66.06_{\pm 1.05}$ | $\mathbf{66.97}_{\pm \mathbf{1.58}}$ | $66.33_{\pm 1.33}$ |

## D.4 OTHER IMBALANCED CL EXPERIMENTS

Due to the page limit, the full results of Table 4 are shown in Table 13. To further explore the performance of VR-MCL in imbalanced settings, we select the more realistic random imbalanced scenario and present the results for a higher imbalance ratio, specifically $\gamma = 5$, in Table 14. For simplicity, we select the SOTA methods from Table 13 as the comparable methods. The results clearly demonstrate that the proposed VR-MCL surpasses other methods in the case of $\gamma = 5$, providing further evidence of the effectiveness of the variance reduction technique in addressing imbalanced CL.

To further demonstrate the effectiveness of the proposed VR-MCL under imbalanced settings, we performed additional verification on various imbalance settings using the Seq-CIFAR100 dataset. The results of these experiments are presented in Table 15.

Table 13: Reduced ResNet-18 performance on the imbalanced Seq-CIFAR10 ($|\mathcal{M}|$=1000) with imbalance ratio $\gamma$=2.

| | $\gamma$ =2 | | $\gamma$ =2 (Reversed) | | $\gamma$ =2 (Random) | |
|---|---|---|---|---|---|---|
| Methods | AAA | ACC | AAA | ACC | AAA | ACC |
| SGD | $36.64_{\pm 0.42}$ | $16.46_{\pm 0.31}$ | $35.57_{\pm 0.70}$ | $17.54_{\pm 0.16}$ | $34.62_{\pm 1.31}$ | $17.33_{\pm 0.34}$ |
| On-EWC | $38.85_{\pm 0.18}$ | $16.92_{\pm 0.31}$ | $37.41_{\pm 0.26}$ | $17.79_{\pm 0.17}$ | $37.26_{\pm 0.49}$ | $14.35_{\pm 1.85}$ |
| A-GEM | $38.50_{\pm 0.25}$ | $17.64_{\pm 0.43}$ | $36.72_{\pm 0.33}$ | $17.43_{\pm 0.47}$ | $37.79_{\pm 0.45}$ | $17.85_{\pm 0.22}$ |
| GEM | $41.29_{\pm 0.42}$ | $17.90_{\pm 0.65}$ | $38.46_{\pm 1.13}$ | $18.37_{\pm 0.25}$ | $39.63_{\pm 1.38}$ | $18.00_{\pm 0.73}$ |
| ER | $52.50_{\pm 0.74}$ | $33.10_{\pm 2.17}$ | $46.58_{\pm 2.29}$ | $28.49_{\pm 3.46}$ | $43.78_{\pm 2.85}$ | $31.10_{\pm 5.38}$ |
| DER | $46.95_{\pm 1.54}$ | $16.82_{\pm 0.98}$ | $40.62_{\pm 1.69}$ | $18.63_{\pm 0.61}$ | $41.44_{\pm 0.65}$ | $18.78_{\pm 0.23}$ |
| DER++ | $62.02_{\pm 0.52}$ | $44.14_{\pm 2.77}$ | $58.22_{\pm 0.85}$ | $39.21_{\pm 3.81}$ | $60.25_{\pm 0.58}$ | $42.83_{\pm 2.60}$ |
| CLSER | $61.37_{\pm 0.69}$ | $47.75_{\pm 0.70}$ | $54.92_{\pm 0.55}$ | $40.51_{\pm 1.15}$ | $56.60_{\pm 1.51}$ | $47.48_{\pm 0.70}$ |
| CBRS | $59.07_{\pm 1.78}$ | $43.81_{\pm 0.13}$ | $57.57_{\pm 1.60}$ | $44.20_{\pm 1.96}$ | $58.16_{\pm 1.57}$ | $44.66_{\pm 3.21}$ |
| Coresets | $61.11_{\pm 1.34}$ | $45.37_{\pm 0.98}$ | $58.12_{\pm 1.37}$ | $45.80_{\pm 1.89}$ | $58.56_{\pm 2.17}$ | $45.63_{\pm 1.89}$ |
| La-MAML | $36.64_{\pm 1.54}$ | $29.17_{\pm 0.91}$ | $32.08_{\pm 2.37}$ | $31.17_{\pm 1.59}$ | $42.52_{\pm 2.81}$ | $31.24_{\pm 2.01}$ |
| VR-MCL | $\mathbf{65.06}_{\pm 0.85}$ | $\mathbf{49.82}_{\pm 1.13}$ | $61.16_{\pm 1.32}$ | $\mathbf{51.36}_{\pm 1.31}$ | $\mathbf{61.91}_{\pm 1.07}$ | $\mathbf{50.74}_{\pm 1.15}$ |

Table 14: Reduced ResNet-18 performance under the random imbalanced scenario with larger imbalance ratio, i.e., $\gamma = 5$ (Random).

| Method | CLS-ER | DER++ | La-MAML | VR-MCL |
|---|---|---|---|---|
| Acc | $35.66_{\pm 1.56}$ | $40.93_{\pm 0.69}$ | $27.46_{\pm 4.65}$ | $\mathbf{41.55}_{\pm \mathbf{3.15}}$ |
| AAA | $53.93_{\pm 0.61}$ | $49.98_{\pm 2.34}$ | $36.85_{\pm 1.33}$ | $\mathbf{54.52}_{\pm \mathbf{1.97}}$ |

Table 15: Reduced ResNet-18 performance on the imbalanced Seq-CIFAR100 ($|\mathcal{M}|$=1000) with imbalance ratio $\gamma$=2.

| Setting | Method | CLSER | DER++ | La-MAML | VR-MCL |
|---|---|---|---|---|---|
| $\gamma = 2$ | Acc | $13.85_{\pm 1.19}$ | $9.19_{\pm 1.20}$ | $11.19_{\pm 1.07}$ | $\mathbf{15.12}_{\pm \mathbf{1.32}}$ |
| | AAA | $23.54_{\pm 1.02}$ | $18.78_{\pm 0.24}$ | $19.54_{\pm 0.38}$ | $\mathbf{24.02}_{\pm \mathbf{1.13}}$ |
| $\gamma = 2$ (Reverse) | Acc | $12.89_{\pm 0.78}$ | $9.66_{\pm 0.03}$ | $11.86_{\pm 0.60}$ | $\mathbf{14.98}_{\pm \mathbf{1.65}}$ |
| | AAA | $18.72_{\pm 1.22}$ | $15.60_{\pm 0.60}$ | $16.04_{\pm 0.53}$ | $\mathbf{20.88}_{\pm \mathbf{1.16}}$ |
| $\gamma = 2$ (Random) | Acc | $13.42_{\pm 1.48}$ | $8.60_{\pm 0.49}$ | $11.32_{\pm 0.91}$ | $\mathbf{14.93}_{\pm \mathbf{0.27}}$ |
| | AAA | $19.10_{\pm 1.89}$ | $15.55_{\pm 0.25}$ | $16.22_{\pm 0.58}$ | $\mathbf{20.13}_{\pm \mathbf{0.57}}$ |

## D.5 OTHER RESULTS USING THE PCCNN BACKBONE

In Table 16, we further investigate the performance gain over the memory buffer size on the small network PcCNN. Combined with the results on reduced ResNet18, it is evident that the proposed VR-MCL has a consistent and significant performance improvement in different buffer sizes and various network structures.

Table 16: Performance of Seq-CIFAR10 with 95% confidence interval on PcCNN. All numbers are the average of 5 runs. $|\mathcal{M}|$ denotes the memory buffer size.

| Method | $|\mathcal{M}|$=200 | | $|\mathcal{M}|$=600 | | $|\mathcal{M}|$=1000 | |
|---|---|---|---|---|---|---|
| | AAA | Acc | AAA | Acc | AAA | Acc |
| SGD | $37.84_{\pm 0.03}$ | $17.58_{\pm 0.20}$ | $37.84_{\pm 0.03}$ | $17.58_{\pm 0.20}$ | $37.84_{\pm 0.03}$ | $17.58_{\pm 0.20}$ |
| On-EWC | $38.72_{\pm 0.58}$ | $17.05_{\pm 0.08}$ | $38.72_{\pm 0.58}$ | $17.05_{\pm 0.08}$ | $38.72_{\pm 0.58}$ | $17.05_{\pm 0.08}$ |
| A-GEM | $41.84_{\pm 0.44}$ | $18.50_{\pm 0.38}$ | $41.54_{\pm 0.14}$ | $18.45_{\pm 0.37}$ | $41.73_{\pm 0.61}$ | $18.78_{\pm 0.38}$ |
| GEM | $45.07_{\pm 0.52}$ | $22.52_{\pm 1.02}$ | $45.11_{\pm 0.34}$ | $22.45_{\pm 0.89}$ | $44.92_{\pm 0.20}$ | $22.10_{\pm 0.60}$ |
| ER | $54.64_{\pm 0.91}$ | $32.46_{\pm 1.58}$ | $58.85_{\pm 0.82}$ | $40.29_{\pm 1.25}$ | $59.26_{\pm 0.69}$ | $41.71_{\pm 1.35}$ |
| DER | $58.06_{\pm 0.18}$ | $38.21_{\pm 0.29}$ | $58.40_{\pm 0.11}$ | $38.46_{\pm 0.45}$ | $57.90_{\pm 0.12}$ | $38.88_{\pm 0.39}$ |
| DER++ | $56.07_{\pm 0.32}$ | $34.28_{\pm 0.79}$ | $62.13_{\pm 0.21}$ | $43.89_{\pm 0.20}$ | $63.55_{\pm 0.16}$ | $47.20_{\pm 0.67}$ |
| CLSER | $58.03_{\pm 0.75}$ | $39.38_{\pm 1.44}$ | $61.53_{\pm 0.32}$ | $44.94_{\pm 1.01}$ | $62.39_{\pm 0.34}$ | $48.24_{\pm 0.85}$ |
| MER | $55.50_{\pm 0.38}$ | $32.01_{\pm 1.22}$ | $63.20_{\pm 0.43}$ | $44.69_{\pm 0.43}$ | $65.75_{\pm 0.63}$ | $51.04_{\pm 0.78}$ |
| La-MAML | $46.36_{\pm 0.98}$ | $29.45_{\pm 0.51}$ | $47.22_{\pm 0.87}$ | $32.78_{\pm 1.53}$ | $47.22_{\pm 0.87}$ | $32.51_{\pm 1.18}$ |
| VR-MCL | $\mathbf{60.88}_{\pm \mathbf{0.22}}$ | $\mathbf{43.41}_{\pm \mathbf{0.30}}$ | $\mathbf{63.86}_{\pm \mathbf{0.52}}$ | $\mathbf{48.85}_{\pm \mathbf{0.66}}$ | $\mathbf{66.02}_{\pm \mathbf{0.31}}$ | $\mathbf{52.67}_{\pm \mathbf{0.25}}$ |

## D.6 OFFLINE CL PERFORMANCE

As shown in Table 17, we also provide the performance of different methods under the offline CL setting where we set the training epochs as 5.

## D.7 INFLUENCE OF THE INNER BATCH SIZE

To investigate the practical implications of the inner batch size, we conducted experiments using various inner batch sizes. The corresponding results are presented in Table 18. The findings indicate that a larger inner batch size consistently leads to improved performance. However, it is worth noting that the performance gains tend to plateau as the inner batch size increases. This can be attributed to the online nature of the setting, where each sample is observed only once. A larger inner batch size reduces the number of outer loop steps, which may potentially compromise the overall performance improvement.

## E ALGORITHM.

As presented in Algo. 1, Steps 1-9 present the update of $\theta$ in the first iteration, where no momentum term is available. Steps 2-6 present the inner loop and steps 7-8 present the estimate of hypergradient in the outer loop. Steps 10-28 represent the update of $\theta$ in $b$-th iteration ($b > 1$). Steps 12-17 are

Table 17: **Offline CL** (i.e., 5 training epochs) performance of Seq-CIFAR10 and longer task sequences Seq-CIFAR100, Seq-TinyImageNet with 95% confidence interval on reduced ResNet-18. The memory buffer size is set as 1000 (i.e., $|\mathcal{M}|$=1000).

| Method | Seq-CIFAR10 AAA | Seq-CIFAR10 Acc | Seq-CIFAR100 AAA | Seq-CIFAR100 Acc | Seq-TinyImageNet AAA | Seq-TinyImageNet Acc |
|---|---|---|---|---|---|---|
| SGD | $42.26_{\pm0.22}$ | $18.87_{\pm0.66}$ | $18.73_{\pm1.67}$ | $7.73_{\pm0.66}$ | $16.89_{\pm0.26}$ | $6.93_{\pm0.03}$ |
| On-EWC | $42.13_{\pm0.18}$ | $20.63_{\pm1.76}$ | $16.99_{\pm2.12}$ | $5.34_{\pm0.42}$ | $14.89_{\pm0.14}$ | $4.85_{\pm0.35}$ |
| IS | $39.84_{\pm0.11}$ | $17.49_{\pm0.49}$ | $15.56_{\pm0.58}$ | $4.98_{\pm0.65}$ | $10.92_{\pm1.44}$ | $2.65_{\pm0.41}$ |
| A-GEM | $42.13_{\pm0.14}$ | $19.12_{\pm0.15}$ | $18.69_{\pm1.81}$ | $7.64_{\pm1.42}$ | $16.73_{\pm0.47}$ | $6.50_{\pm0.35}$ |
| GEM | $48.28_{\pm0.83}$ | $23.75_{\pm3.91}$ | $21.56_{\pm3.57}$ | $10.71_{\pm2.31}$ | $18.12_{\pm0.11}$ | $7.98_{\pm0.33}$ |
| ER | $74.64_{\pm0.76}$ | $63.97_{\pm0.27}$ | $35.37_{\pm2.36}$ | $21.84_{\pm1.75}$ | $26.49_{\pm0.38}$ | $12.47_{\pm0.23}$ |
| DER | $67.38_{\pm1.54}$ | $59.95_{\pm1.96}$ | $18.73_{\pm2.46}$ | $7.67_{\pm1.12}$ | $16.96_{\pm0.44}$ | $6.68_{\pm0.22}$ |
| DER++ | $75.50_{\pm2.02}$ | $67.16_{\pm1.62}$ | $39.15_{\pm0.54}$ | $24.03_{\pm0.38}$ | $25.91_{\pm0.63}$ | $11.54_{\pm0.71}$ |
| CLSER | $75.45_{\pm1.10}$ | $\mathbf{67.36}_{\pm\mathbf{0.27}}$ | $40.18_{\pm1.70}$ | $26.16_{\pm0.34}$ | $26.24_{\pm0.33}$ | $12.98_{\pm0.10}$ |
| ER-OBC | $75.93_{\pm0.95}$ | $65.53_{\pm1.42}$ | $\mathbf{41.15}_{\pm\mathbf{0.53}}$ | $29.02_{\pm0.99}$ | $29.01_{\pm0.93}$ | $16.27_{\pm0.44}$ |
| La-MAML | $60.55_{\pm0.48}$ | $43.58_{\pm0.56}$ | $31.05_{\pm0.28}$ | $19.67_{\pm1.46}$ | $23.07_{\pm0.71}$ | $11.81_{\pm0.55}$ |
| VR-MCL | $\mathbf{77.15}_{\pm\mathbf{0.75}}$ | $67.34_{\pm1.10}$ | $40.02_{\pm0.89}$ | $\mathbf{29.32}_{\pm\mathbf{0.94}}$ | $\mathbf{30.43}_{\pm\mathbf{1.00}}$ | $\mathbf{19.60}_{\pm\mathbf{0.58}}$ |

Table 18: Reduced ResNet-18 performance under different inner batch sizes.

| Inner batch size | 6 | 8 | 10 |
|---|---|---|---|
| Acc | $53.80_{\pm2.36}$ | $56.48_{\pm1.79}$ | $56.81_{\pm1.07}$ |
| AAA | $65.73_{\pm3.06}$ | $66.97_{\pm1.58}$ | $67.26_{\pm0.61}$ |

the inner loops. Steps 18-19 show the computation of the hypergradient for $\theta_b$. In steps 20-25, we perform the inner loop to obtain $\theta'_{b(K)}$ and compute the hypergradient for $\theta_b$ as $\mathbf{g}^{\epsilon_b}_{\theta_{b-1}}$. During the update of memory buffer $\mathcal{M}$ (step 28), we employ the reservoir sampling strategy to ensure that $\mathcal{M}$ is updated in a way that the stored examples are uniformly sampled from the tasks during online training.

### E.1 VARYING TASK LENGTH ANALYSIS

To further demonstrate the effectiveness of the proposed VR-MCL across varying task lengths, we adjust the split mechanism of Seq-CIFAR100 and Seq-TinyImageNet to increase their original task numbers. Specifically, we restructure Seq-CIFAR100, originally composed of 10 tasks with 10 classes each, into 20 tasks, each with 5 classes. Similarly, we modify Seq-TinyImageNet, initially consisting of 20 tasks with 10 classes each, into 40 tasks, each comprising 5 classes. For clarity, we refer to these new division datasets as Seq-CIFAR100$^l$ and Seq-TinyImageNet$^l$ respectively. The performance of various baselines on these two datasets is presented in Table 19.

### E.2 EMPIRICAL RESULTS OF THE OUTER-LOOP LOSS

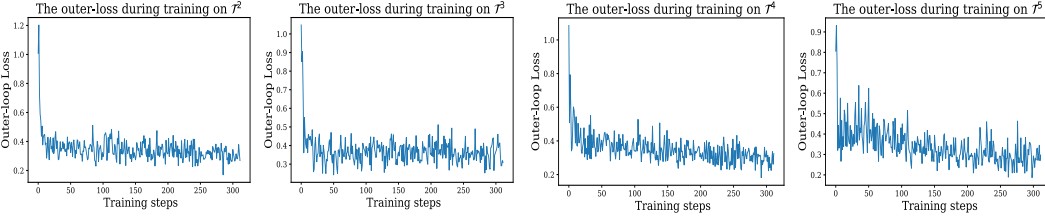

Figure 7: The outer-loop training loss for each task $\mathcal{T}^j$ in Seq-CIFAR10 ($j = 2, 3, 4, 5$).

---

**Algorithm 1** The Algorithm of the proposed VR-MCL.

---

**Require:** Initial model weights $\theta$, initial inner-loop learning rate $\beta$, outer-loop learning rate $\alpha$, number of inner-loop steps $K$, memory buffer $\mathcal{M}$, and the task number in total $N$.

1: Sample training data $\mathcal{T}_{(s)}^1$ $(s=1,...,K)$ from current task $\mathcal{T}^1$.
2: $\theta_{(0)} = \theta, \theta_0 = \theta$
3: **for** $i=1$ to $K$ **do**
4:     obtain samples $\mathcal{T}_{(i)}^1$
5:     $\theta_{(i)} = \theta_{(i-1)} - \beta\nabla\mathcal{L}_{(i)}^1(\theta)$
6: **end for**
7: sample $\epsilon_0$ from $\mathcal{M} \cup \mathcal{T}^1$
8: $\hat{\mathbf{g}}_{\theta_0}^{\epsilon_0} = \nabla_\theta\mathcal{L}(\theta_{(K)}, \epsilon_0)$
9: $\theta_1 = \theta_0 - \alpha\hat{\mathbf{g}}_{\theta_0}^{\epsilon_0}$
10: **for** $j=1$ to $N$ **do**
11:     **for** $b=(j-1)\times B+1$ to $j\times$B **do**
12:         sample training data $\mathcal{T}_{(s)}^j$ $(s=1,...,K)$ from current task $\mathcal{T}^j$.
13:         $\theta_{b(0)} = \theta_b$
14:         **for** $i=1$ to $K$ **do**
15:             obtain samples $\mathcal{T}_{(i)}^j$
16:             $\theta_{b(i)} = \theta_{b(i-1)} - \beta\nabla\mathcal{L}_{(i)}^j(\theta_{b(i-1)})$
17:         **end for**
18:         sample $\epsilon_b$ from $\mathcal{M} \cup \mathcal{T}^j$
19:         $\mathbf{g}_{\theta_b}^{\epsilon_b} = \nabla\mathcal{L}(\theta_{b(K)}, \epsilon_b)$
20:         $\theta_{b(0)}' = \theta_{b-1}$
21:         **for** $i=1$ to $K$ **do**
22:             use the same sampled training data $\mathcal{T}_{(i)}^j$
23:             $\theta_{b(i)}' = \theta_{b(i-1)}' - \beta\nabla\mathcal{L}_{(i)}^j(\theta_{b(i-1)}')$
24:         **end for**
25:         $\mathbf{g}_{\theta_{b-1}}^{\epsilon_b} = \nabla\mathcal{L}(\theta_{b(K)}', \epsilon_b)$
26:         $\hat{\mathbf{g}}_b^{\epsilon_b} = \mathbf{g}_b^{\epsilon_b} + r(\hat{\mathbf{g}}_{b-1}^{\epsilon_{b-1}} - \mathbf{g}_{\theta_{b-1}}^{\epsilon_b})$
27:         $\theta_{b+1} = \theta_b - \alpha\hat{\mathbf{g}}_b^{\epsilon_b}$
28:         Update the memory buffer $\mathcal{M}$ with $\mathcal{T}_{(i)}^j$ using reservoir sampling strategy.
29:     **end for**
30: **end for**

---

Table 19: Experimental results under varying task lengths with $|\mathcal{M}| = 1000$ and a 95% confidence interval. The Notation $N$ denotes the task length.

| Method | Seq-CIFAR-100$^l$ ($N$=20) | | Seq-TinyImageNet$^l$ ($N$=40) | |
|---|---|---|---|---|
| | AAA | Acc | AAA | Acc |
| SGD | $9.23 \pm 0.26$ | $3.34 \pm 0.13$ | $4.95 \pm 0.16$ | $1.19 \pm 0.27$ |
| A-GEM | $10.84 \pm 0.23$ | $3.56 \pm 0.17$ | $6.10 \pm 0.14$ | $1.67 \pm 0.07$ |
| GEM | $16.04 \pm 2.58$ | $7.01 \pm 1.95$ | $7.92 \pm 0.15$ | $2.93 \pm 0.38$ |
| ER | $20.46 \pm 0.48$ | $12.71 \pm 0.28$ | $13.75 \pm 0.12$ | $6.82 \pm 0.11$ |
| DER++ | $16.32 \pm 0.49$ | $8.24 \pm 0.41$ | $9.39 \pm 0.07$ | $3.76 \pm 0.81$ |
| CLSER | $22.03 \pm 0.96$ | $15.39 \pm 2.36$ | $14.93 \pm 0.36$ | $7.74 \pm 0.91$ |
| ER-OBC | $21.04 \pm 0.65$ | $15.87 \pm 1.26$ | $14.92 \pm 0.20$ | $8.45 \pm 2.29$ |
| La-MAML | $17.42 \pm 0.79$ | $10.27 \pm 0.46$ | $10.83 \pm 0.59$ | $5.29 \pm 0.28$ |
| VR-MCL | $\mathbf{24.29} \pm \mathbf{1.07}$ | $\mathbf{17.44} \pm \mathbf{0.97}$ | $\mathbf{18.28} \pm \mathbf{0.14}$ | $\mathbf{10.54} \pm \mathbf{0.30}$ |

Fig. 7 illustrates the value of $\mathcal{L}^{[1:j]}(\theta_{(K)})$ during the training of each task $\mathcal{T}^j$ in Seq-CIFAR10 (j=2,3,4,5). The loss $\mathcal{L}^{[1:j]}(\theta_{(K)})$ exhibits a significant decrease within an average of 5 outer loop update steps, suggesting a close proximity between $\theta_{(K)}$ and $\theta^*$ as stated in Proposition 2.

