# OpenReview forum: "Meta Continual Learning Revisited: Implicitly Enhancing Online Hessian Approximation via Variance Reduction"
_ICLR.cc/2024/Conference — ICLR 2024 oral_

### Official Review · Reviewer_Z5So · 2023-10-30

**Soundness:** 4 excellent
**Presentation:** 4 excellent
**Contribution:** 4 excellent
**Rating:** 10
**Confidence:** 4

**Summary:**

In this paper, the author revisited the methodology of Meta-Continual Learning (Meta-CL) and, for the first time, provides a formal connection between meta-continual learning and seminal regularization-based methods (like Elastic Weight Consolidation (EWC)) which mainly exploits the empirical Hessian matrix to provide the regularization to counter forgetting. The main finding is that Meta-CL methods implicitly utilize the second-order Hessian information through the hypergradient obtained by bi-level optimization for meta-learning. From this new perspective, the author further points out the issue existing in the methodology of Meta-CL, i.e., the presence of erroneous information in the Hessian information due to insufficient memory data. To resolve the problem, the author correspondingly proposes a momentum-based Variance-Reduced Meta-CL (VR-MCL) method and provides extensive theoretical analysis to demonstrate how the proposed method can impose a penalty on the online estimated Hessian such that the model can be updated with caution to preserve crucial parameters. Extensive experiments are conducted on standard continual learning benchmarks, and the proposed theoretical method outperform both representative and state-of-the-art (SOTA) continual learning methods.

**Strengths:**

1. The reviewer really enjoys reading this paper. This should be the first paper that formally and clearly dissects the relationship between seminal regularization-based methods and the methodology of meta-continual learning. The key message and insights are conveyed smoothly in the whole paper, and the author does a really good job of presenting them in a decent way. Table 1 provides a very precise and clear summary and comparison of the seminar and the state-of-the-art regularization-based method for the reader to get their main idea in common, making it easier for the reader to comprehend the novelty and contribution made by the present paper. Figures 1 and 2 are also compact and reduce the difficulties for the reader to understand the technical details of the iterative update process, which also highlights the difference made in this paper.

As the Hessian information is widely used not only in continual learning but also in many different areas of deep learning (e.g., meta-learning and flatness-aware optimization), the reviewer believes that the theoretical findings provided by this paper may not only motivate novel methods on Meta-CL but may also motivate novel methods for other areas in general.

2. The unification of the Meta-CL and Regularization-based method is sound. Although there exist papers that try to unify different regularization-based CL methods in a unified framework, the CL methods they considered are mainly for CL in a fully-supervised setting, to the best of my knowledge, this paper should be the first one to connect the regularization-based CL methods with the methodology of Meta-CL, which may stand as a new research direction in the future.

3. The reviewer also appreciates the understanding provided by the author in Section 4.2 after Proposition 3. It is refreshing to see that the variance-reduce method can ensure cautious updates such that the model can prevent excessive updates triggered by the wrongly estimated low-curvature direction of the Hessian, which may mitigate the partiality and erroneousness in the insufficient memory data, which should also be a desideratum about the kind of model update we should purse for. The insight may also motivate future work in continual learning and may also in areas like parameter-efficient finetuning.

4. The extensive comparison with state-of-the-art methods in both CL and Meta-CL further demonstrates the significance and effectiveness of the proposed method. The questions listed in each subsection of the Experiments section provide good guidance for the reviewer to focus on and reason about the results. It is also great to see that the author also conducts many empirical analyses in both the main paper and supplementary to validate the correctness of the proposed theorem.

**Weaknesses:**

1. In Proposition 3, the author assumes that the batch size for inner step adaptation is sufficiently large. How do we quantify the term "sufficiently large" in reality? Is there any principle we can obtain from the proposed theorem to guide us in choosing the batch size?

**Questions:**

Please refer to the Weaknesses for more details.

---

> ### Author Response · Authors · 2023-11-17
> **Response to Reviewer Z5So**
>
> Thank you sincerely for your thoughtful and positive feedback on our work. We are particularly grateful for your recognition of the various aspects of our research. Below, we have provided a detailed explanation for your remaining concern as follows. Please do not hesitate to let us know if you have any further questions.
>
>  **Q1**: How to quantify the "sufficiently large" inner batch size? Is there any principle we can obtain from the proposed theorem.
>  > - We made this assumption of a sufficiently large batch size for inner step adaptation to (1) **guarantee the accurately estimated inner gradients** given the number of inner steps $K=1$ in Proposition 3, and thus (2) facilitate the proof in Appendix A.3.2.
>  > - In practical applications where $K>1$, as discussed in Appendix A.2, the updating rule incorporates the average of inner gradients over all $K$ steps. This averaging mechanism **alternatively ensures the accuracy of inner gradient estimation, thereby alleviating the necessity for an excessively large batch size**.
>  > - We validate the above analyses with two groups of experiments.
>  >   - We compare the performance of (1) VR-MCL with $K=8$ and an inner batch size of 4 and (2) VR-MCL with $K=4$ and an inner batch size of 8. The results are presented in the table below. The comparable performance of the two cases supports the notion that either a large batch size or a large number of inner steps suffices.
> >
> >| Inner batch size | Case (1)  |        Case (2)           |
> >|:----------------:|:---:|:--------------------:|
> >|   Acc      |  56.16 $\pm$ 2.18   | 56.48 $\pm$ 1.79 |
> >|   AAA      |  66.42 $\pm$ 2.06   | 66.97 $\pm$ 1.58 |
> >
> >    - - On Seq-CIFAR10, we conduct an ablation study of VR-MCL by **fixing the inner steps to $K=4$** and  evaluating two other inner batch sizes of 6 and 10. As anticipated, a larger inner batch size indeed contributes to improved performance, while the improvement brought by increasing the inner batch size tends to plateau. This is attributed to the nature of the online setting, where all samples are seen only once. As a result, a larger inner batch size leads to a reduced number of outer loop steps, potentially compromising the performance gain.
> >
> >| Inner batch size |  6  |          8           | 10  |
> >|:----------------:|:---:|:--------------------:|:---:|
> >|       Acc        | 53.80 $\pm$ 2.36   | 56.48 $\pm$ 1.79 |  56.81 $\pm$ 1.07   |
> >|       AAA        | 65.73 $\pm$ 3.06   | 66.97 $\pm$ 1.58 |  67.26 $\pm$ 0.61  |

---

> > ### Comment · Reviewer_Z5So · 2023-11-19
> >
> > Thanks to the author for the detailed reply. All my concerns have been resolved. As I mentioned in my review comments, this is a good paper that should be highlighted and seen by the community. I have increased my score to reflect this.

---

> > ### Comment · Reviewer_Z5So · 2023-11-22
> >
> > One more comment: Will the author make the official implementation public upon acceptance?

---

> > > ### Author Response · Authors · 2023-11-22
> > > **Response to Reviewer Z5So**
> > >
> > > Thanks for your comment. Yes, we are currently in the process of tidying up the code and we plan to make the official implementation publicly available very soon, upon the paper's acceptance.   Once again, we would like to express our gratitude for your constructive comments and positive feedback on our work.

---

### Official Review · Reviewer_1xci · 2023-10-31

**Soundness:** 3 good
**Presentation:** 3 good
**Contribution:** 3 good
**Rating:** 8
**Confidence:** 3

**Summary:**

This paper focused on the branch of Meta-Continual Learning (Meta-CL) methods in the context of Continual Learning (CL). By characterizing the Meta-CL algorithms as a new perspective of up-to-date Hessian matrix approximation, the authors tried to bridge the gap between the Meta-CL and the regularized-based CL methods. Under this viewpoint, Meta-CL implicitly approximated the Hessian in an online manner through the use of hypergradient in the bi-level optimization process. To address the erroneous information during the Hessian estimation due to the sampling process from the random memory buffer, the authors proposed Variance Reduced Meta-CL (VR-MCL) to control the high variance of the hypergradient under online continual learning. With a theoretical analysis, the authors showed that the proposed VR-MCL is equivalent to the inclusion of a penalty term within the implicit Hessian estimation in Meta-CL. The experimental results on three benchmarks indicated that the proposed method outperformed the regularization-based and Meta-CL baselines.

**Strengths:**

1. The motivation of this work is clear and easy to follow.
2. It is interesting to see that an inherent connection can be built between the regularization-based methods and Meta-CL methods via the roles of the Hessian information in these two methodological streams.
3. This work provided theoretical analyses and empirical verifications to help to better understand the motivation.

**Weaknesses:**

1. Most parts of the mathematical derivations are easy to follow. However, some detailed notations are not clear in the context, which reduces the readability.
2. The motivation of some experimental designs was not too clear, such as the imbalance CL setting.
3. It seems the math derivation process needs some strict assumptions. I doubt the gap between the theoretical findings and the empirical applications.

See the Questions part for more details.

**Questions:**

1. I wonder whether the assumptions during the mathematical derivation always hold in the practical scenarios. For example:
   - In Proposition 2, the authors assumed that $\theta_{(K)}$ is lolcated in the $\epsilon$-neighbourhood of the optimal model parameter. Is it too strong?
   - In Proposition 3, the authors assumed that the batch size of the inner step adaptation is sufficiently large. I wonder how large is enough to make the following analyses hold. And how did the authors set it during the practical training?
2. The motivation for the evaluations under the imbalance CL setting was not clear to me. I did not get the relationship between the superior performance under this setting and the main objective of this paper. Or does the author just intend to show that the proposed method could still perform well under this challenging setting? Besides, it was disappointing to see that the authors did not provide further analyses about why the proposed method could address this challenging setting.
3. In Proposition 2, the authors mentioned the assumption of $\beta$. However, it was not contained in the final main conclusion.
4. After Eqn.(4), $G_{\theta_{b}}$ appeared without further explanations, which made the reader fail to have a straightforward comprehension of the meaning of $\Delta_{b}$.
5. How about the time and memory complexity of the proposed method compared to the baseline approaches, especially the Meta-CL methods, like LA-MAML? Could the authors provide quantitative comparisons? I believe such a comparison will help the readers to better understand the superiority of the proposed VR-MCL.

---

> ### Author Response · Authors · 2023-11-17
> **Response to Reviewer 1xci (1/2)**
>
> We appreciate very much your constructive comments on our paper. Please kindly find our response to your comments below, and all revisions made to the paper are highlighted in blue for your ease of reference. We hope that our response satisfactorily addresses the issues you raised. Please feel free to let us know if you have any additional concerns or questions.
>
>  **Q1**: Whether the assumptions during the mathematical derivation hold in the practical scenarios?
>  > - **The assumption of $\theta_{(K)}$ in Proposition 2.**
>  >   - The optimal model $\theta^*=\arg\min_\theta\mathcal{L}^{[1:j]}(\theta_{(K)})$ within the framework of Meta-CL methods [1], as defined in Proposition 2, is the one that minimizes the loss over all seen tasks in the outer loop, i.e., $\mathcal{T}^1,\mathcal{T}^2,...,\mathcal{T}^j$. Thus, $\theta^*$ is task-dependent, signifying that **the introduction of a new $j$-th task corresponds to a new $\theta^{*}$**.
>  >   - $\theta_{(K)}=U_K(\theta;\mathcal{T}^j)$ as defined in Eqn. (3) represents the parameter at the the $K$-th step optimized toward the current task $\mathcal{T}^j$ in the inner loop.
>  >   - The assumption that $\theta_K$ resides in the $\epsilon$-neighborhood of $\theta^*$ is grounded on our **empirical observation**. Our experiments show that for each $j$-th task, the loss value of  $\arg\min_\theta\mathcal{L}^{[1:j]}(\theta_{(K)})$ becomes small within an average of 5 outer loop steps, indicating a close proximity between $\theta_{(K)}$ and $\theta^*$. Further details regarding this empirical finding are provided in Appendix F.2.
>  >   - We attribute this proximity as the recursive nature inherent in the above optimization process in the context of continual learning. Starting with $\theta^{j-1}$, which minimizes $\mathcal{L}^{[1:j-1]}(\theta_{(K)}^{j-1})$ in the $(j-1)$-th task, $\theta^K$ optimized for task $j$ swiftly approaches the local minimum of $\mathcal{L}^{[1:j]}(\theta_{(K)})$. This fast convergence can be elucidated by decomposing $\mathcal{L}^{[1:j]}(\theta_{(K)})$ into $\mathcal{L}^{[1:j-1]}(\theta_{(K)}) + \mathcal{L}(\theta_{(K)})$, where $\mathcal{L}^{[1:j-1]}(\theta_{(K)}^{j-1})$ has been optimized before.
>  > - **The sufficiently large batch size in Proposition 3.**
>  >   - This assumption of a sufficiently large inner batch size **ensures that the inner gradients of the current task $\mathcal{T}_j$ are highly accurate and stable**, given that Proposition 3 considers the number of inner steps $K$ as 1. Accurate inner gradients facilitate subsequent analyses (i.e.,$\nabla_\theta \mathcal{L}^{j}(\theta_2)\approx \nabla_\theta \overline{\mathcal{L}}^{j}(\theta_{1}) \approx \nabla_\theta \mathcal{L}^{j}(\theta_{1})$), as demonstrated in the proof in Appendix A.3.2.
>  >   - In practice, we adopt multiple inner steps, specifically $K=4$, aligning with the extended version of Proposition 3 (details provided in Appendix A.2). In this version, the prerequisite of Proposition 3, namely accurate inner gradients, is readily fulfilled by averaging over multiple steps.
>  >   - We have further **explored the practical impact of the inner batch size** by evaluating alternative settings, specifically considering inner batch sizes of 6 and 10. From the following table, we conclude that
>  >     - a larger inner batch size indeed yields improved performance, aligning with the aforementioned analysis;
>  >     - the improvement brought by increasing the inner batch size tends to plateau. This is attributed to the nature of the online setting, where all samples are seen only once. As a result, a larger inner batch size leads to a reduced number of outer loop steps, potentially compromising the performance gain.
> >
> > | Inner batch size |  6  |          8           | 10   |
> > |:----------------:|:---:|:--------------------:| --- |
> > |       Acc        | 53.80 $\pm$ 2.36   | 56.48 $\pm$ 1.79 |  56.81 $\pm$ 1.07   |
> >|       AAA        | 65.73 $\pm$ 3.06   | 66.97 $\pm$ 1.58 |  67.26 $\pm$ 0.61  |
> >
>  > [1] Look-ahead meta learning for continual learning. NeurIPS 2020.

---

> ### Author Response · Authors · 2023-11-17
> **Response to Reviewer 1xci (2/2)**
>
> **Q2**: The motivation for the evaluations under the imbalanced CL setting.
> > - As acknowledged by the reviewer in the Summary, our main objectives include addressing "the erroneous information during the Hessian estimation due to the sampling process from the random memory buffer". This is achieved through the proposed VR-MCL, which effectively "controls the high variance of the hypergradient under online continual learning".
> > - In the imbalanced CL setting, a memory buffer $\mathcal{M}$ constructed using the common practice of reservior sampling strategy stores **imbalanced samples across different tasks** [2]. Compared to online continual learning, this imbalance in $\mathcal{M}$ further results in a **less accurate Hessian estimation with higher variance**.
> > -  Thus, we conduct the evaluations under this challenging imbalanced CL setting to specifically validate the effectiveness of VR-MCL, examining whether we achieve our main objective of controlling the high variance. Our results in Table 4 affirmatively confirm this efficacy.
> > - We appreciate the reviewer for pointing this out, and have incorporated this analysis into the main text (Sec. 5 Q3) for better understanding.
>
>  **Q3**: The notation of $\beta$ is not appear in the final main conclusion of the Proposition 2.
>  > To facilitate a transparent comparison between the iterative update rule presented in Proposition 2 and others showcased in Table 1, we adopt $\alpha=\beta^2$ in the main conclusion of Proposition 2. The related proof is provided in Appendix A.2. We appreciate the reviewer for bring this to our attention and have included this clarification in Proposition 2 to enhance readability.
>  >
>
>  **Q4**: $G_{\theta_b}$ needs to be further explained for better understanding of the meaning of $\Delta_b$.
>  >As stated in the main text, $G_{\theta_b}$ represents the true but unknown gradient direction, which corresponds to the **full-batch gradient** calculated over all samples of each task. In online CL, samples for each task are provided in individual batches, making it impossible to obtain all samples of a single task simultaneously and thus rendering $G_{\theta_b}$ unknown. Considering that the gradient $\hat{g}\_{\theta_b}^{\epsilon_b}$ is calculated based on **mini-batch samples**, the difference $\Delta_{b}=\hat{g}\_{\theta_b}^{\epsilon_{b}}-G_{\theta_b}$ can serve as an indicator of the **variance level**.
> >
> > [2] Online continual learning from imbalanced data. ICML 2020.
>
> **Q5**: Time and memory complexity analysis of VR-MCL and other Meta-CL methods.
>  > - We have provided the **time complexity analysis** of VR-MCL and other Meta-CL methods, like MER and La-MAML, in **Appendix D.1. (Table 9)**. For ease of illustration, the results are also summarized in the table below.
>  >   - The analysis clearly demonstrates that VR-MCL significantly reduces training time compared to MER (approximately 93.7% reduction in training time) and requires slightly longer training time compared to La-MAML.
>  >   - To assess the value of the additional training time incurred by VR-MCL, we extended the training epoch of La-MAML from 1 to 2. Despite having an even longer training time than VR-MCL, La-MAML with 2 epochs only achieved a performance of 38.89%, which is extremely lower than the performance of VR-MCL (i.e., 56.48%). This outcome serves as compelling evidence for the superiority of VR-MCL in balancing between training efficiency and performance.
>  >
> > |     Method     | La-MAML | MER |  VR-MCL    |
> > |:--------------:|:--------------:|:--------------:|:---:|
> > |   Training Epochs  |  1   |   1   |  1   |
> > | Training Time (s) |  750.61   |   20697.7   |  1297.10    |
> > | | | | |
> > |   Training Epochs  |  2   |   1   |  1   |
> > | Training Time (s) |  1511.54   |   20697.7   |  1297.10    |
> > | AAA |  53.21%   |   50.99%   |  66.97%   |
> > | Acc |  38.89%   |   36.92%   |  56.48%   |
> >
> > - Regarding **memory complexity**, we compared the memory usage of three methods, including MER, La-MAML, and VR-MCL, using the PcCNN network with $|\mathcal{M}|$=600 when training on Split-CIFAR10. The results are shown in the table below.
> >    - Notably, VR-MCL only requires 8.8% more memory than La-MAML, yet it achieves a performance improvement of 16.07% and 35.3% over Acc and AAA metrics, respectively. This demonstrates the superior "effective" efficiency of VR-MCL.
> >
> > |   Method   | La-MAML |   MER   | VR-MCL  |
> > |:----------:|:-------:|:-------:|:-------:|
> > | Memory(MB) |    1462    |    1306    |    1594    |
> > |    Acc     |   32.78 $\pm$ 1.53      |    44.69 $\pm$ 0.43     |   **48.85** $\pm$ 0.66     |
> > |    AAA     | 47.22 $\pm$ 0.87  | 63.20 $\pm$ 0.43 | **63.86** $\pm$ 0.52 |

---

> > ### Comment · Reviewer_1xci · 2023-11-19
> > **Post-rebuttal comments from reviewer**
> >
> > I appreciate the explanations from the authors. My concerns have been addressed and I don't have further questions. I accordingly increased my score. The additional experimental results during our discussion are encouraged to be added to the final version of this manuscript.

---

> > > ### Author Response · Authors · 2023-11-22
> > > **To Reviewer 1xci**
> > >
> > > We are pleased that the concerns raised by the reviewer have been addressed, and we will incorporate the additional experimental results during our discussion into the final version. Thanks again for the time and effort the reviewer has dedicated to reviewing our paper and providing valuable feedback.

---

### Official Review · Reviewer_6aKN · 2023-10-31

**Soundness:** 3 good
**Presentation:** 3 good
**Contribution:** 3 good
**Rating:** 8
**Confidence:** 5

**Summary:**

The paper introduces a novel approach called VRMCL (Variance Reduced Meta Continual Learning), integrating a hyper-gradient variance reduction technique for Meta Continual Learning (CL). Furthermore, it offers theoretical regret bounds for the proposed method. The paper extensively evaluates the VRMCL method across three datasets, with diverse continual learning scenarios.

**Strengths:**

1. Clarity: The paper is well written and easy to follow.
2. Technical Proficiency: The paper showcases a highly technical.
3. Originality and Novelty: The paper introduces a novel concept focused on diminishing variance in gradient computations concerning memory buffers in online settings.
4. Comprehensive Empirical Validation: The paper includes extensive experiments and comprehensive ablation study which support the claims made in the paper.

**Weaknesses:**

1. Limited Comparison:
    1. While the authors have made comparisons with recent baselines, the paper could benefit from a more extensive comparison by including well-established methods such as FTML[1] and LFW[2]. A broader comparison would provide a more comprehensive evaluation of the proposed method's strengths and weaknesses.
2. Limited Experimental Width:
    1. Although the authors have conducted evaluations on popular datasets like CIFAR10, CIFAR100, and TinyImageNet, it would be good to test the effectiveness of the proposed method on larger datasets, such as ImageNet-1K. This would offer insights into the algorithm's performance in handling catastrophic forgetting in longer sequences.
    2. Additionally, the experiments could be enhanced by varying the number of tasks on each dataset, thereby showcasing the adaptability of VR-MCL under different task configurations.
3. Lack of Memory Update Strategy Explanation:
    1. The paper could benefit from a more thorough explanation of the memory update strategy employed in the VR-MCL algorithm. Given the algorithm's reliance on the Memory Buffer, a clearer and more detailed description of the update mechanism is essential to provide a comprehensive understanding of the methodology.

[1] Finn, C., Rajeswaran, A., Kakade, S., & Levine, S. (2019, May). Online meta-learning. In International Conference on Machine Learning (pp. 1920-1930). PMLR.

[2] Li, Z., & Hoiem, D. (2017). Learning without forgetting. IEEE transactions on pattern analysis and machine intelligence, 40(12), 2935-2947.

**Questions:**

1. Regarding the algorithm, the paper mentions that the memory buffer is updated to ensure a balanced storage of tasks. Could you provide more details on how this task-balancing process is implemented within the algorithm?
2. It would be valuable to include additional experiments as mentioned earlier, especially those assessing the method's performance under scenarios involving varying task lengths across each datasets.

---

> ### Author Response · Authors · 2023-11-17
> **Response to Reviewer  6aKN  (1/2)**
>
> We sincerely thank the reviewer for providing valuable feedback. We detail our response below point by point. Some experimental results have been updated in the revised paper, and any modifications made to the paper are highlighted in blue for your convenience. Please kindly let us know whether you have any further concerns.
>
> **Q1**: More details on the update mechanism of the memory buffer $\mathcal{M}$.
> > - In rehearsal-based CL methods, the memory buffer $\mathcal{M}$ serves as a repository for storing samples from previous tasks. During sequential training, samples are sampled from $\mathcal{M}$ and combined with those from the new task in a joint training process to mitigate forgetting. The proposed VR-MCL **aligns with established rehearsal-based CL methods** [1,2,3] by employing the **reservoir sampling strategy** to update the samples stored in $\mathcal{M}$.
> > - This strategy updates $\mathcal{M}$ to ensure that the stored examples are uniformly sampled from the tasks during online training. Under this updating scenario,
> >   - (1) when the online training tasks have an equal number of samples, the memory buffer $\mathcal{M}$ will contain balanced samples from different tasks.
> >   - (2) if the number of samples varies across tasks, the memory buffer $\mathcal{M}$ will store imbalanced samples across different tasks [4].
> >   - Since the majority of experiments in this paper adhere to the standard setting where all online tasks have an equal number of samples (setting (1) above), we stated that the memory buffer is updated to maintain balanced samples from different tasks.
> >  - Our contributions do not entail a novel update mechanism for the memory buffer. Instead, the proposed variance reduction exactly alleviates inaccurately estimated $\mathbf{H}^j_M$, a potential outcome of imbalanced samples within $\mathcal{M}$ in setting (2).
> >  - Thanks for pointing this out, and in response, we have revised the corresponding expression in Algorithm 1 (Appendix E) and provided additional explanations to eliminate any potential ambiguity.
> >
> > [1] Experience replay for continual learning. NeurIPS 2019.
> >
> > [2] Dark experience for general continual learning: a strong, simple baseline. NeurIPS 2020.
> >
> > [3] Learning fast, learning slow: a general continual learning method based on complementary learning system. ICLR 2021.
> >
> > [4] Online continual learning from imbalanced data. ICML 2020.
>
> **Q2**: A broader comparison between VR-MCL and well-established methods such as FTML [5] and LWF [6].
> > - We **evaluate LWF** [6] on all Seq-CIFAR10, Seq-CIFAR100 and Seq-TinyImageNet datasets. The average results, along with a 95% confidence interval, are provided **in the following table** and have been added into **Table 2 and Table 3 in the main text**. LWF without the use of a memory buffer, akin to other regularization-based methods (e.g., On-EWC) in performance, struggles in this challenging online CL setting.
> >
> > - Regarding FTML [5], we have to adapt it to the online class-incremental continual learning setting we focus on.
> >   - FTML, by design, learns **the model initialization** for each task, adhering to conventional meta-learning. During testing, it **requires awareness of the task identity** to perform fine-tuning on that task from the initialization.
> >   - In the context of online class-incremental continual learning, however, a distinctive challenge arises as **all test samples**, irrespective of whether they are from previous tasks or the current task, are **not provided with the tasks they belong to**.
> >   - Our adaptation of FTML involves (1) considering all test samples from the current task, (2) fine-tuning on the training samples of the current task, and (3) evaluating on all test samples. The results, presented in the table below, advocates the importance of previous endeavors of MER and LaMAML that transform meta-learning into the domain of continual learning settings.
> >   - We have compared with MER and LaMAML, and the results in **Table 2** of the main text showcase that our proposed VR-MCL consistently outperforms both across all three datasets.
> > - Thanks again for the suggestion to compare with the two methods. We will also incorporate these comparisons and discussions in the related work section.
> >
> >| Inner batch size |  Seq-CIFAR10  |      Seq-CIFAR100           | Seq-TinyImageNet  |
> >|:----------------:|:---:|:--------------------:| --- |
> >|LWF (AAA/Acc) | 35.31/18.84  |  11.98/5.63  | 9.21/4.01 |
> >|FTML (AAA/Acc)| 35.21/17.30  |  11.79/5.32  |   8.87/3.29 |
> >|Ours (AAA/Acc)| 66.97/56.48  |  27.01/19.49 |  21.26/13.27 |
> >
> >
> >[5] Online meta-learning. ICML 2019.
> >
> >[6] Learning without forgetting. TPAMI 2017.

---

> ### Author Response · Authors · 2023-11-17
> **Response to Reviewer 6aKN (2/2)**
>
> **Q3**: Additional experiments under scenarios involving varying task lengths across datasets.
> > - First, we would like to humbly emphasize that **our experiments have already covered scenarios with diverse task lengths**. Specifically, the Seq-CIFAR10 dataset contains 5 tasks, Seq-CIFAR100 includes 10 tasks, and Seq-TinyImageNet has 20 tasks. The results demonstrate that VR-MCL consistently surpasses other baselines across the three datasets, notwithstanding the variations in task lengths.
> > - To further address the reviewer's concern regarding catastrophic forgetting under longer task sequences, we conduct additional experiments, as suggested, wherein we **configure each dataset with varying task lengths** to validate the effectiveness of VR-MCL.
> >   - Concretely, we reconfigure Seq-CIFAR100, originally composed of 10 tasks with 10 classes each, into 20 tasks, each encompassing 5 classes. Similarly, we adjust Seq-TinyImageNet, initially consisting of 20 tasks with 10 classes each, into 40 tasks, each comprising 5 classes. For clarity, we hereby denote the two newly split datasets as Seq-CIFAR100$^l$ and Seq-TinyImageNet$^l$.
> >   - We compare the performance of the proposed VR-MCL against ER, GEM, A-GEM, ER-OBC, DER++, La-MAML, and ClsER, with the results listed below (also **included into Appendix F (Table 18)**). Despite a general trend of performance degradation across all methods in the extended task length setting, the proposed VR-MCL **consistently outperforms** the other baselines over both *AAA* and *Acc* metrics.
> >   - The standard practice in the existing continual learning literature [7,8] that involves ImageNet-1k is to divide it into 10 tasks -- a number even fewer than that of Seq-TinyImageNet. We have re-configured ImageNet-1k into 50 tasks with 200 classes each, and will update the corresponding results shortly.
> >
> >|     Method     | Seq-CF100$^l$ | Seq-CF100$^l$ |  Seq-Tiny$^l$    | Seq-Tiny$^l$ |
> >|:--------------:|:--------------:|:--------------:|:---:|:----------------:|
> >|                |  AAA   |   Acc   |  AAA   | Acc        |
> >| SGD |  9.23$\pm$ 0.26   |   3.34$\pm$ 0.13   |  4.95$\pm$ 0.16       | 1.19$\pm$ 0.27  |
> >| A-GEM |  10.84$\pm$ 0.23   |   3.56$\pm$ 0.17   |  6.10$\pm$ 0.14    | 1.67$\pm$ 0.07  |
> >| GEM |  16.04$\pm$ 2.28   |   7.01$\pm$ 1.95   |  7.92$\pm$ 0.15      | 2.93$\pm$ 0.38  |
> >| ER |  20.46$\pm$ 0.48   |   12.71$\pm$ 0.28   |  13.75$\pm$ 0.12     | 6.82$\pm$ 0.11  |
> >| DER++ |  16.32$\pm$ 0.49  |    8.24$\pm$ 0.41    |9.39$\pm$ 0.07     |  3.76$\pm$ 0.81 |
> >| CLSER |  22.03$\pm$ 0.96  |    15.39$\pm$ 2.36   |14.93$\pm$ 0.36    |  7.74$\pm$ 0.91 |
> >| ER-OBC |  21.04$\pm$ 0.65  |    15.87$\pm$ 1.26   |14.92$\pm$ 0.20   |  8.45$\pm$ 2.29 |
> >| La-MAML |  17.42$\pm$ 0.79  |    10.27$\pm$ 0.46  |10.83$\pm$ 0.59   |  5.29$\pm$ 0.28 |
> >| VR-MCL |  **24.29$\pm$ 1.07**  |   **17.44$\pm$ 0.97**   |**18.28$\pm$ 0.14**   |  **10.54$\pm$ 0.30**|
> >
> >[7] How efficient are today's continual learning algorithms? CVPR workshop 2023.
> >[8] A simple but strong baseline for online continual learning: repeated augmented rehearsal. NeurIPS 2022.

---

> > ### Comment · Reviewer_6aKN · 2023-11-22
> >
> > Thanks for the accurate response.
> >
> > My major concerns have been addressed. Overall I feel positive about the work, and I have updated my score accordingly.

---

> > > ### Author Response · Authors · 2023-11-22
> > > **To Reviewer 6aKN**
> > >
> > > We are delighted to see that the major concerns raised by the reviewer have been successfully addressed. We would like to reiterate our deep appreciation for the reviewer's dedicated time and effort in scrutinizing our paper and providing invaluable feedback.

---

### Author Response · Authors · 2023-11-17
**To all Reviewers**

### Summary
>In order to provide greater clarity on the revisions made to our paper and the experiments we conducted to address the reviewers' questions, we have summarized the modifications and experiments made during the rebuttal period as follows:
>
>**Additional Experiments**:
>- We conduct a comparison between the proposed VR-MCL method and two suggested well-established methods, FTML[1] and LWF[2]. The results clearly indicate the superiority of VR-MCL over these two methods across different datasets.  (Reviewer 6aKN Q2)
>- We provide additional experiments under scenarios involving varying task lengths across datasets. The results show that our VR-MCL consistently outperforms the other baselines over *AAA* and *Acc* metrics.   (Reviewer 6aKN Q3)
>- Analysis of the inner batch size. ( Reviewer 1xci Q1 and Reviewer Z5So Q1)
>- The choice of inner update steps K in practice. (Reviewer 1xci Q1)
>- Time and memory complexity analysis. (Reviewer 1xci Q5)
>
>&nbsp;&nbsp;&nbsp;[1] Online meta-learning. ICML 2019.
>
>&nbsp;&nbsp;&nbsp;[2] Learning without forgetting. TPAMI 2017.
>
>
>**Clarification**:
>- More details on the update mechanism of the memory buffer. (Reviewer 6aKN Q1)
>- Illustration of the assumption of $\theta_{(K)}$ in Propsition 2.  (Reviewer 1xci Q1)
>- Illustration of the sufficient large batch size in Proposition 3. (Reviewer 1xci Q1 and Reviewer Z5So Q1)
>- The motivation for the evaluations under the imbalanced CL setting. (Reviewer 1xci Q2)
>- The clarification of $G_{\theta_{b}}$. (Reviewer 1xci Q4)
>
>

---

### Meta-Review · Area_Chair_JCKk · 2023-12-07

**Metareview:**

The paper has received uniformly high ratings from the reviewers, with a consensus that it presents a significant and novel contribution to the field of Continual Learning (CL). Authors first link Meta CL with regularization based methods and later propose a variance reduction method following momentum based variance-reduction for non-convex SGD. Authors also provide a rigorous theoretical analysis and comprehensive empirical validation. I agree with the reviewers that the paper should be accepted and also recommend oral distinction.

**Justification For Why Not Higher Score:**

N/A

**Justification For Why Not Lower Score:**

The paper has no clear weakness, it has a solid theoretical analysis and rigorous empirical analysis. The unification of CL methods using bilevel optimization framework is also quite interesting. Considering the wide interest in the topic of CL, and the quality of the method, I think the paper should be shared with the community as an oral talk.

---

### Decision · Program_Chairs · 2024-01-16

Accept (oral)